# Reporter CRISPR screens decipher *cis*-regulatory and *trans*-regulatory principles at the Xist locus

Till Schwämmle[1], Gemma Noviello[1], Eleni Kanata[1,2], Jonathan J. Froehlich[1], Melissa Bothe[1], Alexandra Martitz[1,2], Aybuge Altay[3], Jade Scouarnec[1], Vivi-Yun Feng[1,2], Heleen Mallie[1], Martin Vingron[3] & Edda G. Schulz[1] ✉

Developmental genes are controlled by an ensemble of *cis*-acting regulatory elements (REs), which in turn respond to multiple *trans*-acting transcription factors (TFs). Understanding how a *cis*-regulatory landscape integrates information from many dynamically expressed TFs has remained a challenge. Here we develop a combined CRISPR screening approach using endogenous RNA and RE reporters as readouts. Applied to the murine *Xist* locus, which is crucial for X-chromosome inactivation in females, this method allows us to comprehensively identify Xist-controlling TFs and map their TF–RE wiring. We find a group of transiently upregulated TFs, including ZIC3, that regulate proximal REs, driving the binary activation of Xist expression. These basal activators are more highly expressed in cells with two X chromosomes, potentially governing female-specific Xist upregulation. A second set of developmental TFs that include OTX2 is upregulated later during differentiation and targets distal REs. This regulatory axis is crucial to achieve high levels of Xist RNA, which is necessary for X-chromosome inactivation. OCT4 emerges as the strongest activator overall, regulating both proximal and distal elements. Our findings support a model for developmental gene regulation, in which factors targeting proximal REs drive binary on–off decisions, whereas factors interacting with distal REs control the transcription output.

Precise and robust control of developmental genes is achieved through large and complex *cis*-regulatory landscapes[1]. In mammals, developmental genes are driven by the combined activity of up to 20 regulatory elements (REs) per gene and tissue[2–4]. Activity of each RE in turn is determined by multiple *trans*-acting signals in the form of sequence-specific DNA-binding transcription factors (TFs)[5]. Understanding how TF–RE wiring decodes multiple parallel inputs to control gene expression requires systematic approaches to map TF–RE interactions at scale.

Current high-throughput methods rely on TF-binding or sequence motifs but cannot assess functional regulatory relationships[6–9]. Pooled CRISPR screens represent a highly scalable approach to map functional TF–gene interactions[10]; however, they cannot identify the RE that mediates the interaction. To close this gap, we have developed a CRISPR screen variant that uses an RE reporter for phenotypic enrichment to functionally associate TFs with REs in a systematic manner.

We use our approach to investigate the regulation of *Xist*, an essential developmental gene with a tightly controlled and well-defined expression pattern. To trigger inactivation of one X chromosome in females, *Xist* integrates information on developmental stage and sex of the embryo. This ensures its upregulation in a female (XX)-specific and monoallelic fashion[11]. The *Xist* locus has been extensively studied for several decades, yet our understanding of how it decodes

[1]Systems Epigenetics, Otto Warburg Laboratories, Max Planck Institute for Molecular Genetics, Berlin, Germany. [2]Department of Biology, Chemistry, Pharmacy, Freie Universität Berlin, Berlin, Germany. [3]Department of Computational Molecular Biology, Max Planck Institute For Molecular Genetics, Berlin, Germany. ✉e-mail: edda.schulz@molgen.mpg.de

information remains limited. To identify Xist-controlling REs, we have recently performed a noncoding CRISPR screen at the onset of random X-chromosome inactivation (XCI). The screen detected a comprehensive set of 26 REs involved in initial Xist upregulation, many of which are associated with the Xist-regulating long noncoding RNA genes *Tsix*, *Jpx*, *Ftx* and *Xert*[12]. Using chromatin state as a proxy for RE activity, we observed that distal elements exclusively respond to developmental state, whereas promoter-proximal REs were affected by X-chromosome number. This finding suggests distinct functional roles for different RE sets. Which *trans*-acting factors drive these developmental and XX-specific activation patterns remain incompletely understood.

In mice, random XCI is established during the transition from the naive to the formative pluripotent state[13–15]. Whereas several pluripotency factors (NANOG, OCT4 and ZFP42/REX1) have been shown to repress Xist in the naive state[16–20], developmental Xist activators remain largely unidentified. XX-specific Xist upregulation is generally believed to be driven by X-encoded Xist regulators that act in *trans*[21,22]. Two such X-linked Xist activators have been identified, RLIM/RNF12 and Jpx[23–25]. However, even their combinatorial, heterozygous deletion failed to abrogate Xist expression[26]. Therefore, additional X-dosage mediators are yet to be identified.

To understand how information on developmental stage and sex of the embryo is integrated at the *Xist* locus, we set out to systematically identify Xist regulators at the onset of XCI. We find a large set of previously unknown Xist activators through a pooled CRISPR interference (CRISPRi) screen, including the X-linked TF ZIC3 and the formative master regulator OTX2. Using our CRISPR screen variant that combines CRISPRi screens with a reporter assay, we systematically map TF–RE interactions across activating elements in Xist's *cis*-regulatory landscape. We show that promoter-proximal REs are controlled by ZIC3 and a group of autosomal TFs with XX-biased expression. We propose that this regulatory axis governs initial Xist upregulation in a binary fashion to ensure inactivation of one X chromosome in females. A second group of factors, including OTX2, controls Xist expression independent of sex by interacting with distal REs. This group of activators is primarily required to achieve high transcript levels, which we show to be important for efficient X-chromosome silencing.

## Results

### Identifying Xist-controlling TFs through CRISPRi screens

To identify factors that might regulate Xist expression in a sex-specific and developmental fashion, we performed a pooled CRISPRi screen. For all expressed TF genes, we tested whether their depletion would affect Xist upregulation at the onset of XCI. We used differentiating mouse embryonic stem cells (mES cells) grown in serum and 2i/LIF-containing medium (2iSL), differentiated for 2 days by 2i/LIF (2iL) withdrawal, which closely resembles the developmental trajectory in vivo, as discussed below[12,27,28]. To induce gene repression, we used a split dCas9–KRAB system, stably expressed in female mES cells (TX1072 SP107), where the KRAB repressor domain is tethered to dCas9 in response to abscisic acid (ABA) (Fig. 1a, top left). The CRISPR library (TFi Lib) encompassed 11,058 single guide RNAs (sgRNAs) targeting 570 expressed TF genes and 32 non-TF controls previously implicated in Xist regulation (Extended Data Fig. 1a–d and Supplementary Note 3). Of these, we treat 17 factors for which a loss of function has been shown to affect Xist expression as high-confidence controls (Supplementary Table 1).

After lentiviral transduction of the TFi Lib, cells are differentiated, stained for Xist RNA using a FlowFISH assay and sorted into populations with no (Xist^Neg), low (Xist^Low) or high (Xist^High) expression (Fig. 1a and Extended Data Fig. 1e–g). We sorted two different Xist-positive populations (Xist^+) to be able to distinguish factors that are involved in binary control of Xist expression ('basal activators', identified in Xist^Low versus Xist^Neg and in Xist^High versus Xist^Neg) and those required for high RNA levels ('boosting factors', detected only in Xist^High versus Xist^Neg). For

an orthogonal validation of the screen results, we performed a second, smaller screen using a different CRISPRi system and a slightly adapted screen setup (Fig. 1a, bottom, Extended Data Fig. 1h–i and Methods). We made use of our recently developed degron-controlled dCas9–HDAC4 CasTuner[29] system and only sorted the Xist^High and Xist^Neg populations. Both screens displayed the expected quality control metrics (Extended Data Figs. 2 and 3 and Supplementary Note 1) and correlated well (Pearson correlation $r = 0.8$; Extended Data Fig. 3h–i).

The screens identified many known Xist regulators (for example, RNF12, Jpx, Ftx and RIF1) and other controls (Fig. 1b–d, black, and Supplementary Note 2). Surprisingly, OCT4 (POU5F1) was identified as the strongest Xist activator, despite previous reports that classified the TF as an Xist repressor[16,18]. In total, we identified 39 previously unknown putative Xist regulators, including 24 activators and 15 repressors (Fig. 1d and Supplementary Table 1).

Among the activators, 13 TFs were classified as basal regulators (enriched in Xist^High and Xist^Low populations), which potentially control Xist in a binary fashion (Fig. 1d). A set of 11 TFs were identified as boosting factors (exclusively detected in the Xist^High population), suggesting that they primarily drive high Xist expression levels (Fig. 1d). The strongest activators in the first group included OCT4, the X-linked TF ZIC3 and the INO80 subunit NFRKB, whereas the strongest regulator in the boosting group was the epiblast regulator OTX2 (Fig. 1d)[16,30–32].

### OCT4 acts as an Xist activator during differentiation

As OCT4 was previously described as a repressor under different differentiation conditions or in undifferentiated cells[16,18], we characterized its role in Xist regulation in more detail. We used our CasTuner cell line to repress *Oct4* in two undifferentiated (2iSL, SL) and two differentiating (−2iL, EpiLC) conditions (Fig. 2a, Extended Data Fig. 4a,b). Xist was upregulated 13–93-fold in undifferentiated mES cells following *Oct4* knockdown (Fig. 2b–d and Extended Data Fig. 4c,d), confirming a repressive role of OCT4 under these conditions[16]. Repression of *Oct4* during differentiation, by contrast, led to reduced Xist levels in both tested conditions, supporting an activating role (Fig. 2b–d and Extended Data Fig. 4c,d). OCT4 binds several distal enhancer elements of Xist in the Xert region, specifically upon differentiation (EpiLC, Fig. 2e)[12,30]. We, thus, propose that OCT4 acts as a direct activator of Xist during differentiation. Although we cannot exclude a direct repressive effect in undifferentiated cells, repression might potentially be indirect. Oct4 depletion has previously been reported to drive dedifferentiation toward trophectoderm, which is associated with upregulation of several GATA TFs that we recently identified as potent Xist activators[29,33,34].

### Xist activators are transiently upregulated at XCI onset

Next, we aimed to systematically dissect how a broader set of Xist regulators were modulated by X-chromosome number and developmental state. We performed a high-resolution RNA sequencing (RNA-seq) time-course experiment to assess global expression dynamics in XX and XO cells during the first 96 h of 2iL withdrawal (Fig. 3a and Supplementary Table 2). Our differentiation protocol captured the developmental trajectory between embryonic days E3.5 and E6.5, as demonstrated by coembedding our dataset with a published in vivo single-cell (sc)RNA-seq time course (Fig. 3b and Extended Data Fig. 5a)[35,36].

To assess the expression dynamics of the identified Xist regulators, we grouped all TF genes targeted by the TFi Lib according to their expression patterns (naive, transient or committed) (Fig. 3c,d). As expected, Xist repressors were most strongly enriched in the naive cluster ($P = 0.21$, Fisher's exact test; Fig. 3e and Extended Data Fig. 5b), which is downregulated when Xist is upregulated (Fig. 3c,d). Xist activators, on the other hand, were almost exclusively present in the three transient clusters ($P = 0.05–0.36$; Fig. 3d,e).

Notably, most basal activators (enriched in Xist^High and Xist^Low), including the top-scoring basal Xist regulators OCT4, ZIC3 and

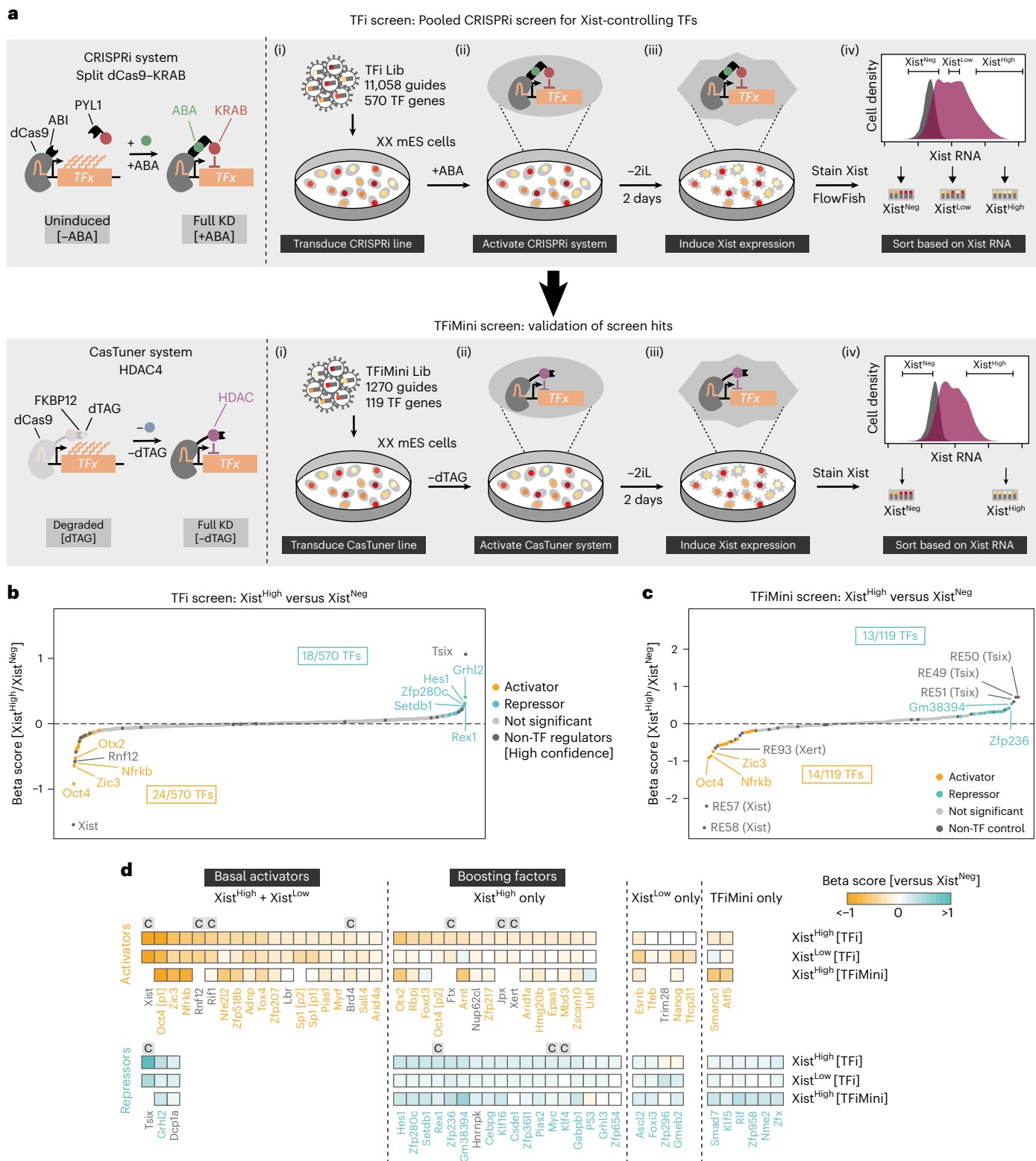

**Fig. 1 | A CRISPR screen targeting expressed TFs to identify Xist regulators.**
**a**, Schematic of the CRISPRi systems (left) and experimental workflow (right) to identify *trans*-regulators of Xist in the TFi (top) and TFiMini (bottom) screens. KD, knockdown. **b,c**, Comparison of Xist^High and Xist^Neg populations in the TFi (**b**) and TFiMini screens (**c**). Significantly enriched or depleted target genes are colored in teal or orange, respectively (MAGeCK mle, Wald $P \leq 0.05$). Non-TF controls are colored in dark gray (only high confidence, as indicated in Supplementary Table 1).

**d**, Heat map depicting Xist regulators identified by comparing Xist^High or Xist^Low to the Xist^Neg population in the TFi and the TFiMini screen (MAGeCK mle, Wald $P \leq 0.05$). Orange and teal gene names indicate TFs that activate and repress Xist, respectively. Non-TF controls are labeled in black and high-confidence controls are marked with a 'C'. Targets with empty positions in the bottom row were not included in the TFiMini screen.

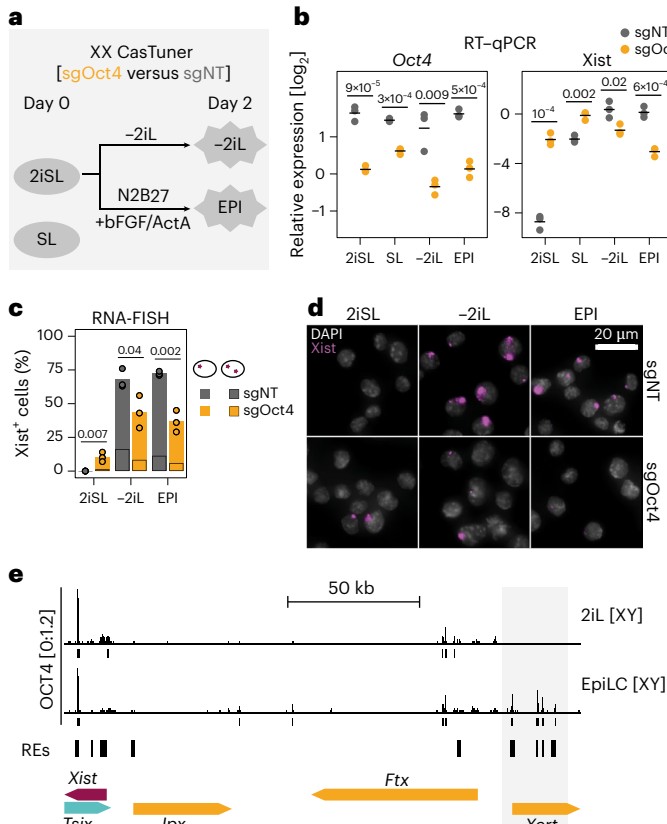

**Fig. 2 | OCT4 acts as an Xist activator during differentiation. a**, Experimental setup used in **b**–**d**. Knockdown of *Oct4* was induced 2 days before differentiation through dTAG removal. bFGF, basic FGF; ActA, activin A. **b**–**d**, Effect of *Oct4* knockdown on Xist and *Oct4* expression, assessed by reverse transcription (RT)–qPCR (**b**) and RNA-FISH (**c**,**d**). A total of 100 cells were counted manually per condition in **c** and representative pictures are shown in **d**. The experiments were performed in three biological replicates. *P* values of an unpaired two-sided *t*-test are indicated in the plot. **e**, Top: published ChIP-seq data, depicting binding of OCT4 at the Xist locus before and during differentiation in XY mES cells[30]. Bottom: RE track shows all Xist-controlling REs in the region[12].

NFRKB, were part of the earliest transient cluster (Transient 1, maximal expression at 17 h; Fig. 3f, yellow), which peaked before Xist upregulation. Boosting factors, by contrast, were mostly found in the later transient clusters 2 (maximal expression at 30 h) and 3 (maximal expression at 36 h) (Fig. 3f, yellow). To systematically assess which factors were affected by differentiation and X-chromosome number, we performed an analysis of variance (ANOVA) (Fig. 3f, red, and Extended Data Fig. 5c). Of 26 identified Xist activators, 18 responded significantly to differentiation, whereas nine were affected by X-chromosome number.

Interestingly, the majority of activators in the earliest cluster (transient 1) exhibited XX-biased expression, whereas the regulators in the later clusters (transient 2 and 3) did not (Fig. 3f,g and Extended Data Fig. 5d). This pattern was also reflected in the two activator groups from the TFi screen; basal activators were expressed in an XX-biased manner, whereas most boosting factors were not (Fig. 3h,i and Extended Data Fig. 5d). These trends were confirmed in vivo and in a different dataset (Extended Data Fig. 5e,f). Taken together, we found that Xist activators are transiently upregulated at the onset of XCI. Basal activators identified in the screens were generally upregulated early (before Xist upregulation) and often exhibited XX-biased expression. These factors might, thus, contribute to female-specific Xist upregulation. Boosting factors were upregulated slightly later and responded mostly to differentiation cues.

## Reporter screens uncover functional TF–RE interactions

In the next step, we aimed to dissect how the REs within the *Xist* locus responded to the TFs we identified. We built upon a comprehensive set of REs that regulate Xist, which we previously identified through a noncoding CRISPRi screen[12]. To comprehensively map TF–RE wiring, we developed a CRISPR screen variant using fluorescent RE reporters as screen readout.

We generated RE reporter cell lines for 21 Xist-controlling REs in our XX CRISPRi mES cell line (Fig. 4a,b and Extended Data Fig. 6a). These included 12 activating REs located either close to the *Xist* promoter (proximal) or up to several 100 kb upstream (distal), nine repressive elements[12] and a control line without RE (noRE). RE sequences were inserted into random genomic positions upstream of an *Fgf4* minimal promoter and a GFP reporter (Fig. 4a).

To verify that the reporter lines recapitulated the activity of the endogenous REs, we performed a differentiation time course (Fig. 4c and Extended Data Fig. 6b,c) and assessed the correlation of fluorescence intensity with endogenous RE activity, assessed through accessibility (assay for transposase accessible chromatin with high-throughput sequencing (ATAC-seq)) and H3K27ac (cleavage under targets and tagmentation (CUT&Tag)). We decided to focus on activating elements, as the correlation was high (Pearson correlation coefficient, $r$ = 0.2–0.6; Extended Data Fig. 6d,e)[12]. These contained four promoter-proximal REs, including the Xist transcription start site (TSS) RE58 and three segments of the RE57 element within *Xist*'s first exon (RE57L/M/R), as well as seven distal REs, including the *Jpx* promoter RE61, the *Ftx*-associated RE85, the Xert-related REs 93 and 95–97 and the *Rnf12* promoter RE127 (Fig. 4b,c)[12]. Similar to the endogenous locus, the proximal REs were already active before differentiation, whereas the activity of the distal REs increased throughout the time course (Fig. 4c).

To map TF–RE wiring, we performed a CRISPRi screen for each reporter using the split dCas9–KRAB system and the TFiMini sgRNA library (Fig. 4d, Extended Data Figs. 7 and 8a, Supplementary Note 1, and Supplementary Table 3). Notably, seven factors, including the strong Xist activator NFRKB, significantly affected the noRE reporter, probably by regulating the minimal *Fgf4* promoter (Extended Data Fig. 8b), and had to be removed from the downstream analyses.

To quantify functional interactions, we defined an interaction score. We computed the $z$ score of the log$_2$-transformed GFP$^{High}$-to-GFP$^{Low}$ sgRNA ratios and normalized them to the values obtained in the noRE control screen (Supplementary Note 3). In total, the analysis revealed 166 functional TF–RE interactions (false discovery rate (FDR) ≤ 0.2, one-sample *t*-test; Fig. 4e). The number of interactions per RE (5–27) and per TF (1–10) were highly variable (Extended Data Fig. 8c,d). The direction of the identified interactions (activating or repressive) typically matched the results of the TFi screen (Fig. 4f). Moreover, the presence of the cognate TF-binding motif within the RE was associated with an increased interaction score for high-confidence Xist activators (Extended Data Fig. 8e). This suggests that potent Xist activators act in part through direct binding.

High reporter activity appeared to increase screen sensitivity, as the number of identified interactions per RE and the correlation between TF and reporter screen results were increased for high-intensity reporters (Extended Data Fig. 8f,g). Supporting this, principal component analysis (PCA) revealed that only five high-activity reporters were well separated from the noRE control: a proximal cluster containing REs 57L/M and a distal cluster consisting of REs 61, 85 and 96 (Fig. 4g). All of these REs were most strongly activated by OCT4, suggesting that the TF acts as a master regulator of the Xist locus during random XCI (Fig. 4e). Moreover, ZIC3 broadly interacted with the indicated reporters but most strongly with the proximal REs 57M/L. In contrast, OTX2 was specifically enriched for the distal RE96.

To assess whether the TFs interacting with specific REs have distinct functional roles, we integrated reporter and TFi screen data. We

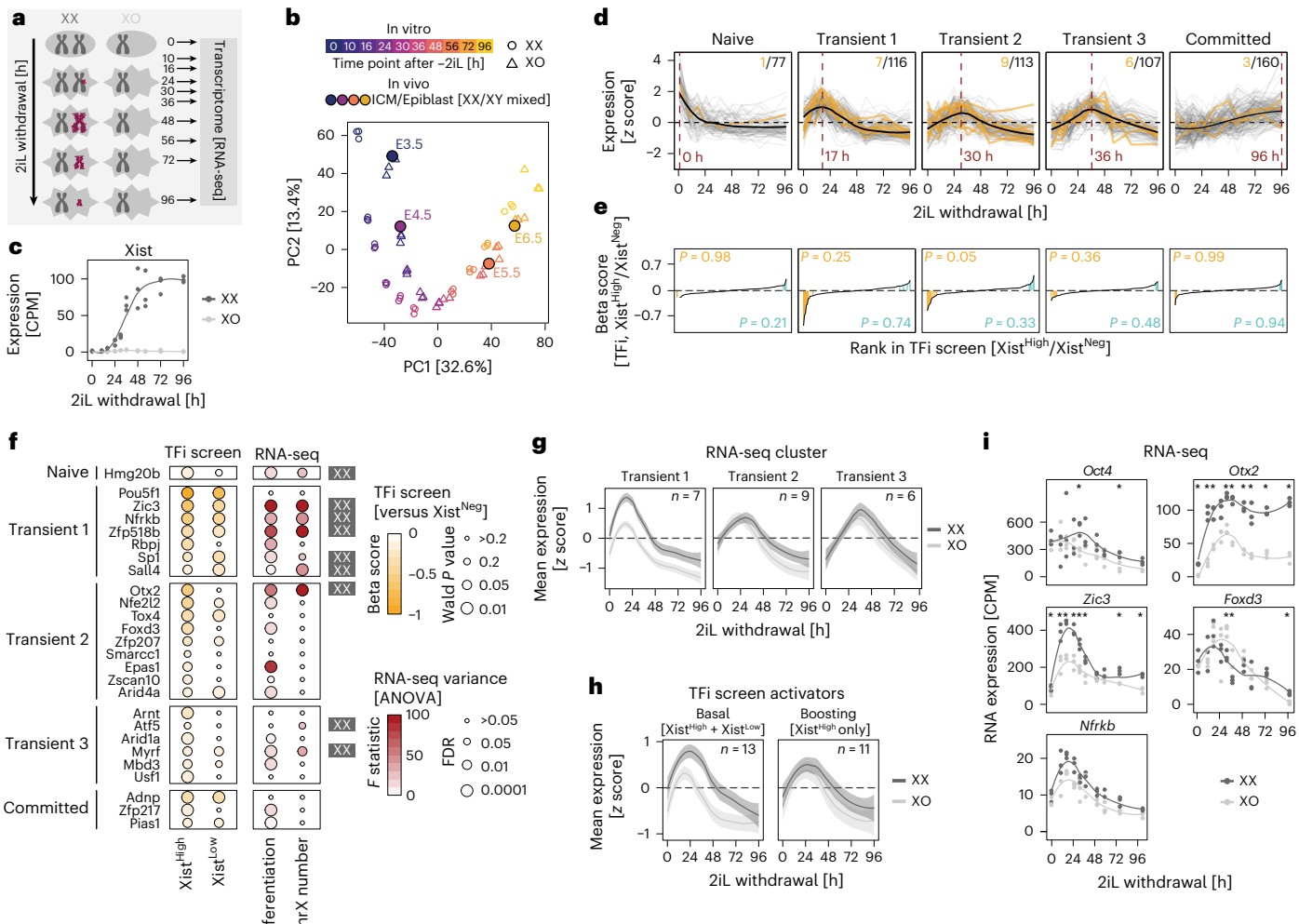

**Fig. 3 | Different groups of Xist activators respond to X-chromosome number and differentiation cues. a**, Experimental setup used in **b**–**i**. The experiment was performed in three biological replicates. Xist RNA upregulation and spreading along the X chromosome (burgundy) and X-chromosome activity (chromosome size) are indicated. **b**, Joint embedding of the RNA-seq time course (open symbols) with sex-mixed in vivo scRNA-seq data of mouse embryos aggregated at four developmental time points (E3.5–E6.5) in pseudobulk (filled circles)[35]. **c**, Xist expression dynamics. CPM, counts per million. **d**, Clustering of z-score-transformed XX expression dynamics for all TF genes included in the TFi screen (details in Supplementary Note 3). Xist activators are shown in orange. The number of genes and activators per cluster (top right) and the time point of maximal expression levels (purple dashed line) are indicated. **e**, Ranked plot depicting TFi screen results (beta score) for all TFs in each cluster. Activators (orange) and repressors (teal) are indicated and enrichment for each class was tested with a one-sided Fisher's exact test (P value indicated). **f**, Dot plot

summarizing the TFi screen results and RNA-seq analysis for all Xist activators identified in the TFi and TFiMini screens. Left: statistics (MAGeCK mle, Wald P value) and effect size (beta score) for the Xist$^{High}$ versus Xist$^{Neg}$ and Xist$^{Low}$ versus Xist$^{Neg}$ comparisons (yellow). Right: contribution of differentiation or X-chromosome number to the expression variance in the RNA-seq time course (two-way ANOVA) (red). The directionality for the X-chromosome effect is indicated next to the plot (FDR ≤ 0.05). **g**,**h**, The z-score-transformed RNA-seq expression dynamics of Xist activators in XX and XO cells grouped according to RNA-seq analysis (**g**; transient clusters) or TFi screen results (**h**). A smoothed average across all genes in the group (line) and a 95% confidence interval (shaded backdrop) are shown. **i**, RNA-seq dynamics of selected Xist activators. Differential expression between XX and XO cells is indicated by a black asterisk (DESeq2, Wald FDR < 0.05)[53]. In **c**,**i**, the lines represent a smoothed average of three replicates across all time points.

ranked TFs on their TFi screen score in the Xist$^{Low}$ (basal activation) or Xist$^{High}$ (boosting factors) populations and plotted cumulative interaction scores for each individual RE (Fig. 4h). Distal REs preferentially interacted with boosting factors, whereas proximal REs responded to both groups. This suggests that the factors controlling distal REs tend to boost Xist levels, whereas those acting on proximal elements also control basal Xist activation.

In the next step, we investigated whether TFs interacting with a specific RE would exhibit distinct expression dynamics. Gene set enrichment analysis (GSEA) showed that interactors of the proximal REs 57L/M were enriched in the naive and transient 1 RNA-seq expression clusters (Fig. 4i). Activating interactions for REs 61, 85 and 96, on the other hand, were enriched for TF genes in the transient 2–3 groups.

Importantly, proximal RE activators showed XX-biased expression, whereas distal RE activators did not (Fig. 4i and Extended Data Fig. 8h–j). This suggests that Xist is regulated in two steps. Initial binary control is mediated by proximal REs in a sex-biased manner, whereas a subsequent boost to high expression levels relies on distal elements.

## Distinct TFs control Xist activation and expression levels

Our global analyses revealed a pattern where basal activation appeared to be associated with early regulators acting through proximal elements, whereas boosting factors were upregulated later and acted through distal enhancers. To test this emerging model, we chose two early basal activators (NFRKB and ZIC3) and two late boosting factors (OTX2 and FOXD3) for in-depth analysis (Fig. 5a). CasTuner-mediated

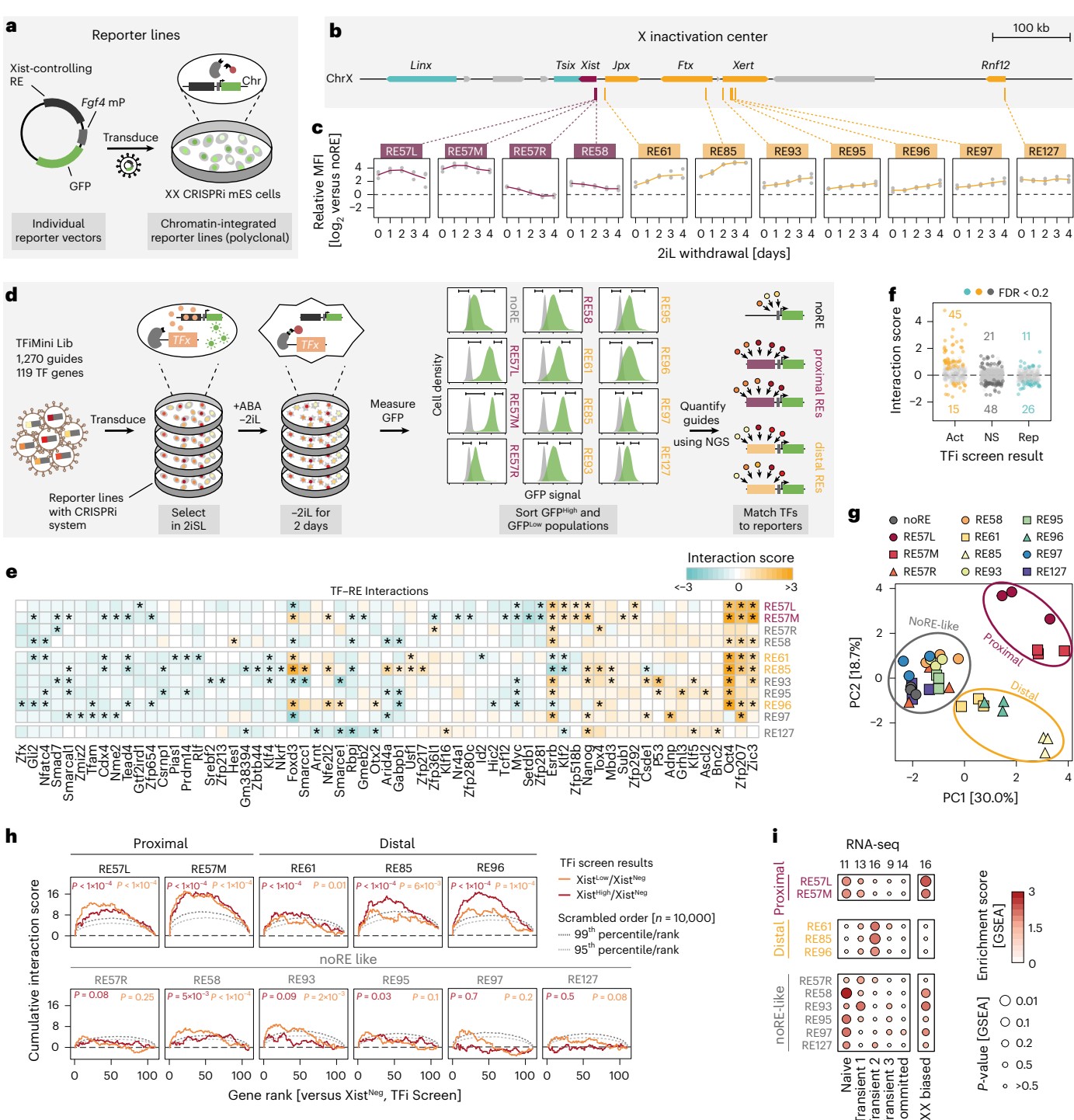

**Fig. 4 | Reporter screens identify functional TF–RE interactions. a**, Schematic outline of the genomically integrated reporter constructs. **b**, The genomic landscape around the Xist locus with Xist regulators shown in teal (repressive) or orange (activating). **c**, Reporter expression dynamics for selected proximal (burgundy) and distal REs (orange). Relative mean fluorescence intensity (MFI) represents the GFP signal in the reporter line normalized to the noRE control. The mean (line) of $n = 3$ biological replicates (dots) is shown. **d**, Schematic outline of the reporter screens: after TFiMini library transduction, induction of the CRISPRi system (ABA) and differentiation (−2iL), 12 reporter lines are sorted into GFP[High] (top 10%) and GFP[Low] (bottom 10%) populations and guide frequencies are quantified by NGS. The screens were performed in three biological replicates. **e**, Heat map displaying interaction scores derived from the reporter screens. Only TF targets with at least one significant interaction (asterisks; Benjamini–Hochberg-corrected two-sided one-sample $t$-test, FDR ≤ 0.2) are shown. Exact $P$ values are provided in Supplementary Table 3. **f**, Interaction scores for all assessed TF–RE combinations, with TFs grouped by effect on Xist expression

(Act, activator; Rep, repressor) in the TFi screen (Xist[High] versus Xist[Neg]). Significant interactions (as in **e**) are colored and their number is indicated. NS, not significant. **g**, PCA of the $\log_2$-transformed fold changes of all TF genes between the GFP[High] and GFP[Low] populations per reporter line and replicate. Groups returned by $k$-means clustering are circled. **h**, Cumulative interaction scores for selected reporter screens ordered by the results of the TFi screen. TF genes are ranked from strongest activator to strongest repressor in the Xist[High] versus Xist[Neg] (red) and Xist[Low] versus Xist[Neg] (orange) comparisons. An empirical $P$ value, calculated by comparing the mean score of the cumulative distribution to scrambled rankings ($n = 10,000$ bootstrap samples), is depicted. The 99th (dark gray) and 95th (light gray) percentiles of the bootstrap samples are shown as dotted lines. **i**, Left: dot plot depicting GSEA results, investigating enrichment for expression groups from the RNA-seq time course in the reporter screens (Fig. 3). Right: results for the set of XX-biased factors (Fig. 3; ANOVA, FDR ≤ 0.05). Only TFs with at least one significant TF–RE interaction are included in the analysis. The number of TFs in each group is shown above the plot.

knockdown achieved 80–95% efficiency and resulted in 2.5–5.5-fold Xist reduction, with minimal cross-regulation between factors (Extended Data Fig. 9a–c). CUT&Tag profiling of active chromatin marks upon knockdown revealed distinct regulatory patterns (Fig. 5b,c and Supplementary Table 4). Whereas knockdown of ZIC3 and NFRKB reduced H3K27ac and H3K4me3 at the Xist promoter, the distal REs were only weakly affected (Fig. 5b,c). Conversely, knockdown of FOXD3 and OTX2 had no effect on the Xist promoter but induced a large reduction in H3K4me1 and H3K27ac at the distal elements (Fig. 5b,c). Interestingly, the knockdown affected all Xert elements (REs 93 and 95–97), whereas the reporter screens only identified interactions with RE96 (Fig. 4e), suggesting crosstalk between REs at the endogenous locus. Nevertheless, our analysis confirmed that the early regulators ZIC3 and NFRKB indeed primarily affect proximal elements, whereas the late factors OTX2 and FOXD3 modulate distal regions.

To assess whether the observed interactions are mediated by direct binding, we reanalyzed public chromatin immunoprecipitation sequencing (ChIP-seq) datasets for ZIC3, OTX2 and FOXD3 in differentiating (male) mES cells (Extended Data Fig. 9d,e). We detected no FOXD3 binding at the Xist locus, suggesting indirect regulation. ZIC3 and OTX2 showed binding to proximal (RE57) and distal (REs 93 and 95–97) elements, respectively, in general agreement with our results. However, OTX2 occupied all XertE REs (Extended Data Fig. 9e) but interacted only with RE96 in the reporter screen (Fig. 4e), suggesting again crosstalk between neighboring REs. Given that some of the regulatory interactions we observed appear to be indirect, we analyzed genome-wide chromatin changes upon knockdown to examine effects on cell state (Extended Data Fig. 9f–h). Whereas NFRKB and ZIC3 had smaller effects (1,000–3,000 peaks affected), FOXD3 knockdown caused widespread changes (8,000–15,000 peaks affected) and impaired differentiation[37], explaining its indirect regulation of Xist.

To test whether the selected factors have functionally distinct roles in Xist regulation, we assessed Xist expression upon knockdown at the single-cell level through RNA fluorescence in situ hybridization (FISH). Through automated image analysis, we quantified the fraction of Xist-expressing cells (frequency) and the amount of Xist RNA per chromosome (intensity). Whereas all four factors reduced both frequency and intensity, the relative contributions of the two modalities to the total knockdown effect were variable (Fig. 5d–g, Extended Data Fig. 9i and Supplementary Table 4). The early factors ZIC3 and NFRKB had a more pronounced effect on frequency compared to the late factors, confirming a role in basal Xist activation. Signal intensity was affected most strongly by the late factor OTX2, whereas the knockdown of the early factor NFRKB had minimal effects.

These analyses suggest that basal Xist activation and high expression levels can be controlled independently from each other. Basal activators control initial upregulation, potentially dependent on X-chromosome number. Subsequently, high expression levels are obtained by activation of distal REs within *Jpx*, *Ftx* and *Xert* by a second wave of late TFs, which include OTX2.

### Distal REs are required for efficient silencing during XCI

As our analyses suggested that high Xist expression levels are controlled by a dedicated regulatory axis, we speculated that they might be required for complete Xist-mediated gene silencing. To test this hypothesis we used a homozygous ΔFtx–Xert deletion line, which lacks most of the distal REs that boost Xist expression[12]. This deletion reduces Xist levels ~2–3-fold, with a minimal effect on Xist+ cell frequency[12], resembling the effect of targeting the same REs by CRISPRi (Extended Data Fig. 10a–d). To quantify chromosome-wide Xist-mediated silencing, we performed allele-specific scRNA-seq for the ΔFtx–Xert line and a wild-type control on day 4 of differentiation, analyzing 502 cells with a median coverage of 55 X-chromosomal genes per cell (Fig. 6a,b, Extended Data Fig. 10e–g and Supplementary Table 5). As expected, the fraction of Xist+ cells was slightly reduced in the

ΔFtx–Xert deletion line (63.5% versus 45.4%; Fig. 6c), whereas Xist levels within Xist+ cells were diminished (fold change of the median = 2.01; Fig. 6d). We identified the inactive X (Xi) in each cell (Supplementary Note 3) and calculated the fraction of reads arising from the Xi (chrX allelic fraction) as a measure of silencing efficiency (Fig. 6e and Extended Data Fig. 10h,i). Compared to wild-type cells, silencing was significantly attenuated in the deletion line (median: 0.43 versus 0.30; Fig. 6e).

To directly test whether reduced Xist levels cause silencing defects, we performed a second scRNA-seq experiment, titrating Xist levels on day 4 of differentiation using our CasTuner system, targeting the Xist TSS or distal REs (Fig. 6f, Extended Data Fig. 10j–o and Supplementary Table 5). We captured 10,993 cells across multiple dTAG concentrations, generating a range of Xist expression levels that correlated with silencing efficiency (Fig. 6g–i). Binning cells by Xist expression revealed a dose-dependent relationship between Xist levels and silencing, with high sensitivity at the low Xist levels observed in the ΔFtx–Xert mutant (Fig. 6j–l, pink).

Individual gene analysis revealed variable dose sensitivity. We fitted dose–response curves and used the median effective dose ($ED_{50}$) parameter, reflecting the normalized Xist level at which allelic expression is reduced by 50%, as a measure of sensitivity (Fig. 6j). Whereas the median $ED_{50}$ was 0.46, indicating that most genes are partially silenced at wild-type Xist levels, some required higher levels for silencing ($ED_{50} > 1$) (Fig. 6m–o and Extended Data Fig. 10p,q). The latter tended to be silenced more slowly in a previous time-course analysis[38] and were located further from Xist on the X-chromosome (Extended Data Fig. 10r–u).

By titrating Xist levels, we showed that silencing increases gradually with increased expression, without a strict threshold. Analysis of a mutant lacking distal REs revealed that the resulting reduced Xist RNA levels lead to a silencing defect. Our results, thus, showed that the activity of distal REs is critical for efficient X-chromosomal silencing across a cell population.

## Discussion

Here, we used a combination of pooled CRISPR screens to systematically characterize input decoding by an entire developmental gene locus. First, we comprehensively identified the *trans*-regulators of Xist at the onset of random XCI using traditional CRISPRi screens and detected a set of previously unknown regulators. We then linked the identified TFs to Xist-controlling REs through several CRISPR screens with fluorescent RE reporters as the phenotypic readout. By integrating the screen results with TF gene expression patterns, we showed that Xist is activated during formative pluripotency in a two-step process, where two functionally distinct groups of Xist regulators are sensed by different RE classes. Whereas high Xist levels are driven by boosting factors through distal enhancers, initial Xist upregulation is governed by a set of XX-biased basal activators that primarily control promoter-proximal REs (Fig. 7).

Several of the previously unknown Xist activators we identified, including ZIC3, OTX2 and FOXD3, have been implicated in the exit from the pluripotent state[30,31,39]. This suggests a tight coupling of this developmental transition and XCI establishment. OTX2 and FOXD3 are essential for embryonic development, as their absence results in embryonic lethality around gastrulation[40,41], whereas absence of ZIC3 results in nonlethal developmental defects[42].

Basal activators are expressed in an XX-biased manner before initial Xist activation, potentially contributing to female-specific expression. One of these factors is ZIC3, a TF encoded on the X chromosome. Similarly to RNF12, ZIC3 might act as a dose-sensitive X-linked Xist activator, proposed to drive female specificity of Xist expression[21,22]. ZIC3 binds the proximal RE57 in close proximity to the established Xist regulators YY1 and REX1 and may, thus, cooperate with them to establish the correct Xist expression pattern[19,20,31].

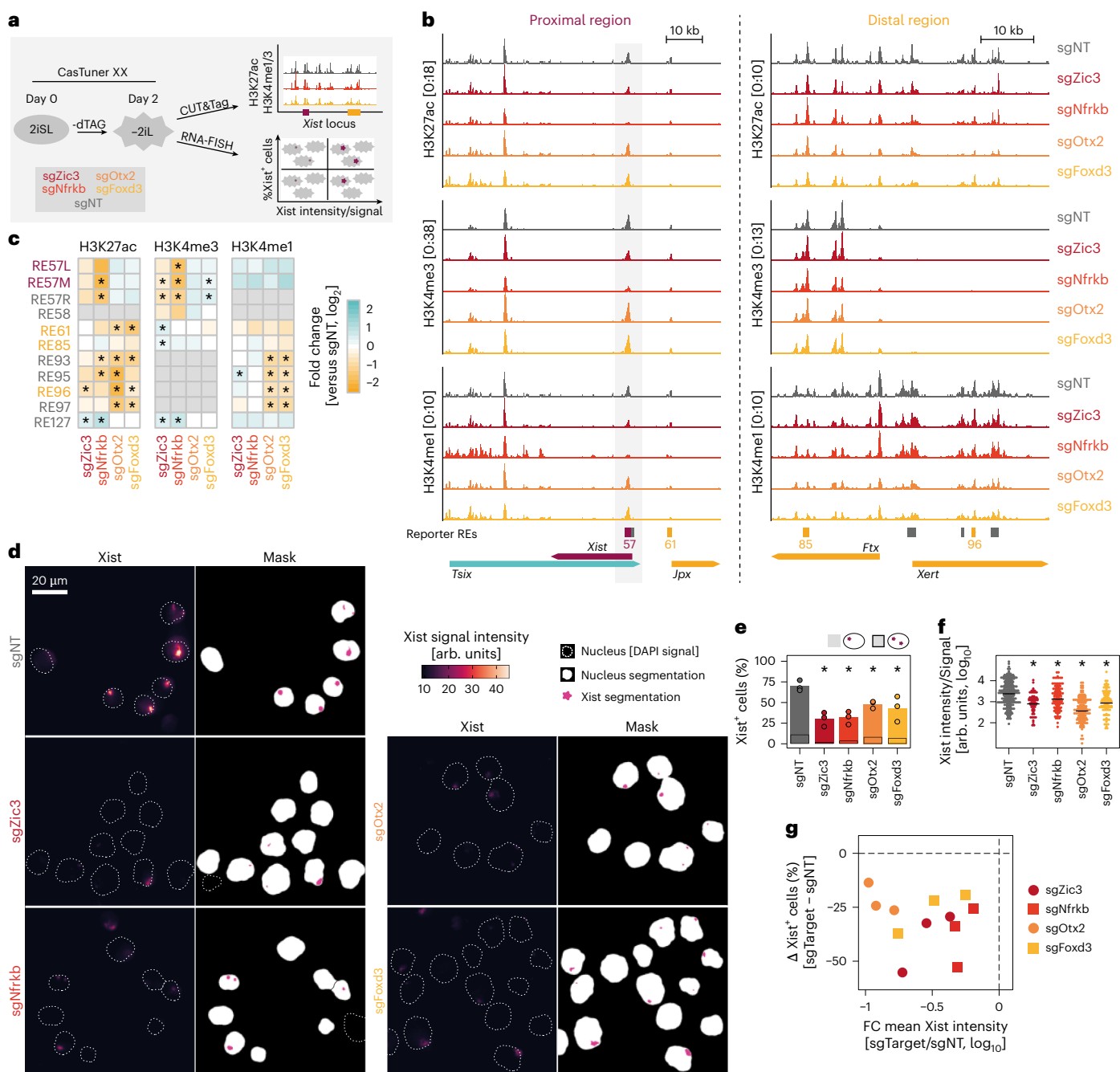

**Fig. 5 | Basal Xist activation and full transcript levels are governed by distinct mechanisms. a**, Schematic outline depicting the experimental setup: CasTuner lines targeting Zic3, Nfrkb, Otx2 and Foxd3 were differentiated for 2 days and Xist RNA signals and activity of Xist-controlling REs were assayed by RNA-FISH (**d**–**g**) and CUT&Tag (**b**–**c**), respectively. **b,c**, Active histone marks following knockdown of different Xist activators. In **b**, the genomic regions around the *Xist* promoter (left) and around enhancers (right) are shown. Three biological replicates were merged for visualization. In **c**, the signal at the REs assayed in the reporter screens in Fig. 4 is quantified. REs with <5 average counts were excluded (gray) and results from a two-sided unpaired *t*-test are shown (asterisks; $P \leq 0.05$). **d**–**f**, RNA-FISH for Xist upon knockdown of Xist activators, segmented through automated image analysis. Example images and the associated segmentation are shown in **d**. The percentage of Xist⁺ cells (**e**) and the signal intensity within Xist signals (**f**) were quantified. In **e**, the mean (bar) of $n = 3$ biological replicates (circles) is shown. Significance was assessed with a two-sided paired *t*-test (asterisk; $P \leq 0.05$). In **f**, the replicates were merged while maintaining an equal number of cells per replicate and significance compared to the sgNT control is marked by a black asterisk (two-sided rank-sum Wilcoxon test, $P \leq 0.05$). The exact $P$ values are 0.03, 0.03, 0.04 and 0.04 in **e** and $6 \times 10^{-30}$, $7 \times 10^{-10}$, $4 \times 10^{-5}$ and $9 \times 10^{-11}$ in **f** for Zic3, Nfrkb, Otx2 and Foxd3, respectively. **g**, Differences in Xist⁺ cells (**e**) and mean Xist intensity (**f**) following TF gene knockdown relative to the sgNT control are shown for individual replicates. FC, fold change.

---

Boosting factors maximize Xist expression by preferentially interacting with the distal Xist REs within *Jpx*, *Ftx* and *Xert*, which are insensitive to X-chromosome number[12]. Accordingly, their expression levels are unaffected by the number of X chromosomes in the cell. Boosting factors are mostly dispensable for basal Xist activation but

drive high expression levels, which we showed guarantees efficient X inactivation, in agreement with recent findings that Xist RNA levels determine the extent of gene silencing[43–45]. Whether additional factors maintain Xist expression at later time points remains an open question, as our screens were performed during XCI initiation. Their

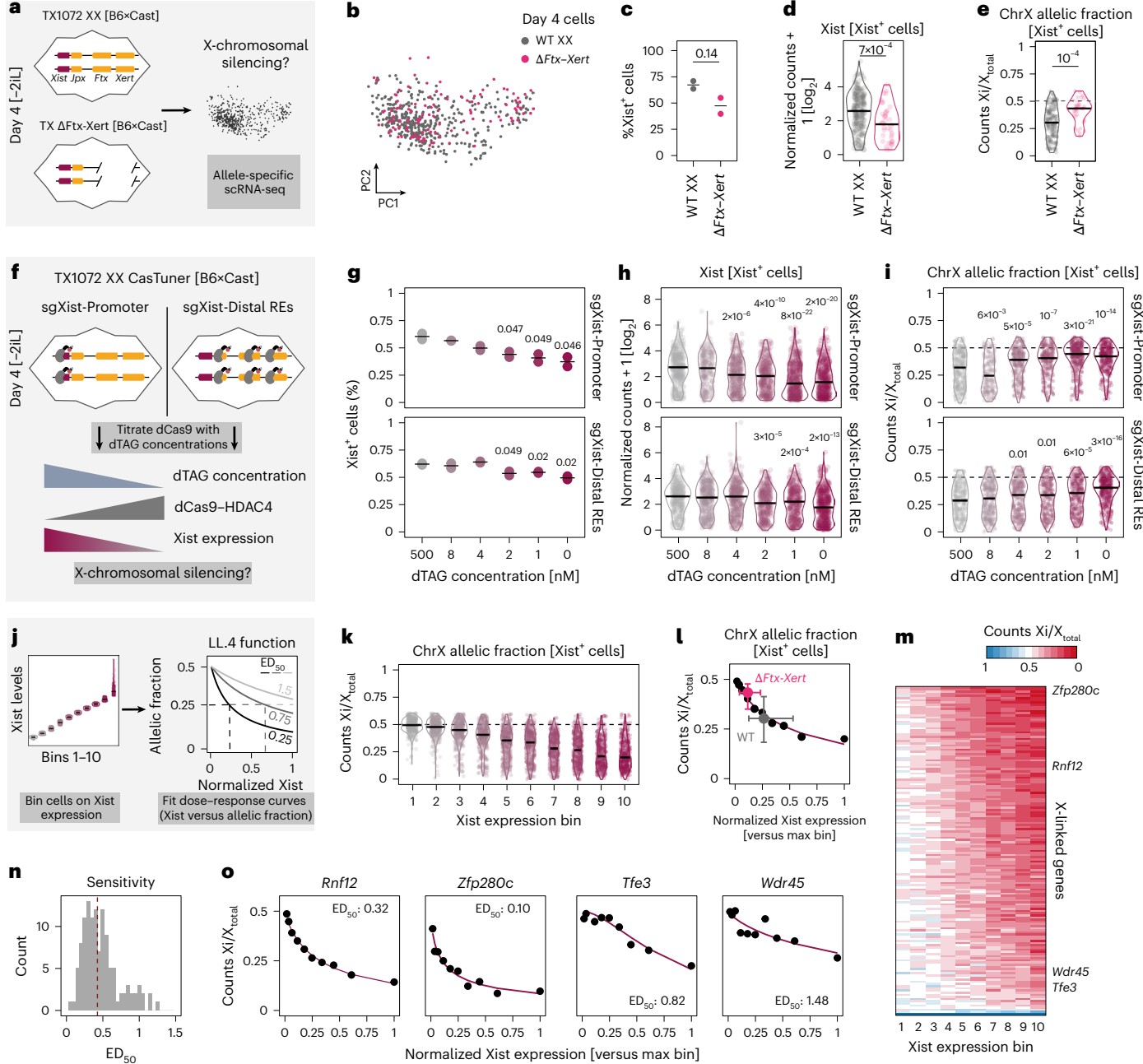

**Fig. 6 | Distal REs are required for efficient silencing during random XCI.**
**a**, Experimental setup used in **b**–**e**. **b**, PCA depicting the transcriptomes of wild-type (WT) and Δ*Ftx–Xert* cells. **c**, Percentage of Xist⁺ cells (>0 counts). The mean (line) of *n* = 2 replicates (dots) is shown. Significance was assessed using a two-sided unpaired *t*-test (*P* ≤ 0.05). **d**,**e**, Xist levels (**d**) and ChrX allelic fraction (**e**) within Xist⁺ cells. Replicates were merged, resulting in 250 and 49 wild-type and Δ*Ftx–Xert* cells in **d** and 175 and 32 wild-type and Δ*Ftx–Xert* cells in **e**, with the median indicated as a black line. The *P* value of a two-sided rank-sum Wilcoxon test is indicated. **f**, Experimental outline used in **g**–**o**, where Xist levels are titrated with CasTuner, with guides targeting the *Xist* TSS (sgXist-Promoter) or REs 61, 85, 93 and 96 (sgXist-Distal REs). **g**–**i**, As in **c**–**e**, but for the Xist titration experiment: percentage of Xist⁺ cells (**g**), Xist levels (**h**) and ChrX allelic fraction in Xist⁺ cells (**i**). Only *P* values < 0.05 for comparison with the 500 nM dTAG condition are

indicated in the plots. **j**, Schematic outline of the dose–response analysis of the Xist titration experiment. Monoallelic Xist⁺ cells were grouped according to Xist expression and a four-parameter log-logistic function was used to estimate for each gene the relative Xist level, where allelic expression is reduced by 50% (ED₅₀). **k**, As in **e**,**i**, but for the Xist expression bins. **l**, Dose–response analysis comparing allelic fraction of the entire X chromosome to normalized Xist expression (dots), fitted using a four-parameter log-logistic function (line). The median values (big dots) and 25th and 75th percentiles (error bars) are shown for the WT (gray) and the Δ*Ftx–Xert* deletion line (pink) from **a**–**e**. **m**, Allelic fraction of individual genes across the Xist expression bins. **n**, Histogram of ED₅₀ values for individual genes, excluding escapees (121 genes). The median ED₅₀ (0.46) is indicated (dashed line). **o**, Example dose–response curves for individual genes.

transient upregulation might function to ensure particularly high Xist levels to initiate silencing, whereas lower levels might be sufficient to maintain XCI.

In our proposed model, Xist upregulation occurs in two distinct stages, each controlled by different activator groups. Our model

parallels a previous study, where two TFs at different developmental stages work together to ensure lineage-specific expression of the *lsy-6* gene in *Caenorhabditis elegans*[46]. An initial 'priming' step by the early activator keeps the locus accessible and sensitive to a later regulator that drives high transcript levels. This mirrors our finding that basal

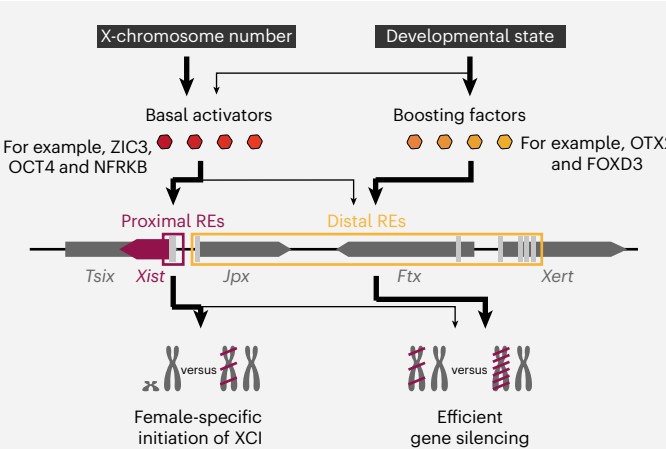

**Fig. 7 | Signal integration at the *Xist* locus.** Schematic depicting the regulatory logic at the Xist locus during random XCI. The X-chromosome number drives female-specific initiation of XCI through the activation of proximal REs by basal activators. The developmental state is additionally sensed by distal REs through boosting factors and promotes high Xist levels that are required for efficient gene silencing.

activators primarily promote an active chromatin state of the *Xist* promoter region. One such activator, NFRKB, like the established Xist activator YY1, functions as a subunit of the INO80 chromatin remodeling complex[20,32], likely maintaining promoter accessibility. A similar association of proximal elements with binary gene control was described for the *Rex1* gene in mice[47]. It is expressed in pluripotent cells but later silenced by promoter DNA methylation, making it insensitive to distal REs. Another example is the *Pitx1* locus during limb development, where expression is reduced by 35–50% upon deletion of the distal enhancer *Pen*, whereas the number of PITX1-positive cells is only decreased by 8–17%, supporting a primary role for *Pen* in boosting expression levels[48,49]. It will be an intriguing question for future studies whether the principle that basal activation and transcriptional boosting are mediated by proximal and distal regulatory regions, respectively, applies more broadly to developmental gene regulation. Moreover, are TFs specialized for binary or gradual control and do they then preferentially bind proximal and distal REs, respectively?

In our study, we developed an approach to investigate developmental gene regulation in a locus-wide fashion. Our reporter screens present a valuable tool for assaying single loci, where the number of regions of interest is limited. We assayed 1,210 TF–RE combinations and detected a total of 166 interactions. Importantly, these regulatory links represent functional interactions and not correlations or TF binding but include indirect interactions. The in-depth characterization of a single *cis*-regulatory landscape also enables a direct comparison of RE characteristics within the endogenous locus and within an ectopically inserted reporter cassette. At the *Xert* locus, TF interactions diverge between the two conditions. Whereas OTX2 knockdown affects four distal REs (93 and 95–97) endogenously, only RE96 responds in reporters, suggesting hierarchical regulation where RE96 activates neighboring enhancers.

This regulatory relationship is reminiscent of the *Pen* enhancer at the *Pitx1* locus and of facilitator elements recently described at the α-globin locus, which can potentiate nearby enhancers[48,50]. A possible approach to further scale up TF–RE mapping is the use of amplicon RNA-seq as a phenotypic readout instead of reporter fluorescence. Whereas the first steps toward that goal have recently been taken, the sensitivity and reliability of this approach remain to be investigated[51]. It might also overcome the challenge to identify regulators of REs that drive only low reporter activity. In our assay, the number of identified interactions increased with the reporter strength. It remains

unclear whether this observation stems from technical limitations or from inherent biological differences, where stronger REs are generally regulated by more TFs. Lastly, our approach identifies functional interactions but cannot easily distinguish direct from indirect effects for individual TF–RE links. This limitation could be alleviated through systematic integration of information on TF binding. Recent studies developed a massively parallel binding assay and a multiplexed version of ChIP-seq (ChIP-DIP) that might fill that gap in the future[9,52].

## Online content

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

## Methods

### Cell lines

The female mES cell line TX1072 (clone A3) is an $F_1$ hybrid cross between C57BL/6 (B6) and CAST/EiJ (Cast) mouse strains. It carries a doxycycline-inducible promoter in front of the Xist gene on the B6 allele and an rtTA insertion at the *Rosa26* locus[54]. The TX1072 XO line (clone B7) has lost the B6 X chromosome and is trisomic for chromosome 16 (ref. 55). The TX Δ*Ftx–Xert* line (clone F9) carries a homozygous deletion spanning the activating REs in *Ftx* and *Xert* (RE85–RE97)[12]. The female TX SP107 cell line (clone B6) stably expresses a split CRISPRi system, consisting of PYL1-KRAB-IRES-Blast and ABI-tagBFP-SpdCas9. Dimerization of the PYL1 and ABI domains was induced by the addition of 100 μM ABA to the cell culture medium 24 h before differentiation[12]. The female TX SP427 (clone B2) carries the CasTuner CRISPRi system, based on an HDAC4 domain fused with the conditional degron domain FKBP12-F36V. The cell line was generated by *PiggyBac*-mediated transposition of the TX1072 A3 line with the pSLPB2B-FKBP12_F36V-hHDAC4-SpdCas9-tagBFP-PGK-Blast plasmid (Addgene, 187956)[29]. To ensure continuous degradation of the dCas9–HDAC4 fusion protein, the cell line was cultured in the presence of 500 nM dTAG-13 (Tocris). The system was induced by removal of dTAG-13 from the medium 48 h before differentiation. The Lenti-X 293T cell line (Takara) was used for lentiviral packaging. All ES cell lines were karyotyped by shallow DNA sequencing, which confirmed a correct karyotype unless otherwise specified. The sources of all cell lines are listed in Supplementary Table 6.

### mES cell culture and differentiation

TX1072 mES cells and derived mutant lines were cultured on 0.1% gelatin-coated flasks in serum-containing medium supplemented with 2i and LIF (2iSL conditions) (DMEM (Sigma), 15% ES cell-grade FBS (Gibco), 0.1 mM β-mercaptoethanol, 1,000 U per ml LIF (Millipore), 3 mM GSK3 inhibitor CT-99021 and 1 mM MEK inhibitor PD0325901 (Axon)). The cells were cultured at a density of $4 \times 10^4$ cells per cm² and passaged every second day. If not stated otherwise, differentiation was induced at a density of $2.1 \times 10^4$ cells per cm² on 10 μg ml⁻¹ fibronectin (Merck or Corning) through 2i/LIF withdrawal (−2iL conditions) (DMEM, 10% FBS (Gibco) and 0.1 mM β-mercaptoethanol). For the CRISPR screens, differentiation was induced at a density of $3.6 \times 10^4$ cells per cm². To differentiate towards EpiLC fate, cells were cultured in serum-free N2B27 medium (50% neurobasal medium (Gibco), 50% DMEM/F12 (Gibco), 1× GlutaMAX (Thermo Fisher), 0.1 mM β-mercaptoethanol (Sigma), 0.5× N2 (Thermo Fisher) and 0.5× B27 (Thermo Fisher)), supplemented with 20 ng ml⁻¹ activin A (StemCell) and 12 ng ml⁻¹ fibroblast growth factor 2 (FGF2; StemCell). RNA-FISH using intronic probes of the X-linked gene *Huwe1* was regularly performed before experiments to confirm that cells had retained two X chromosomes.

### Xist titration

To generate multiple population with variable Xist expression levels, the female TX SP427 (clone B2), carrying the CasTuner CRISPRi system, was transduced with a multiguide plasmid targeting either the Xist TSS region (sgXist-Promoter) or the distal REs 61, 85, 93 and 96 (sgXist-Distal REs). To achieve gradual depletion of the dCas9–HDAC4 fusion protein, the cells were cultured in the presence of six different dTAG-13 (Tocris) concentrations: 0, 1, 2, 4, 8 and 500 nM.

### Cloning multiguide plasmids

For individual CRISPRi experiments, 3–4 different guides targeting gene promoters and distal REs (Supplementary Table 6) were introduced into an sgRNA expression plasmid (SP199) by Golden Gate cloning, as described previously[56]. Following cloning, each guide was controlled by a different Pol III promoter (hu6, mu6, hH1 or 7sk). To this end, gene blocks carrying mu6, hH1 or 7sk promoter sequences fused to an optimized sgRNA constant region (Integrated DNA Technologies)

were amplified with primers containing part of the guide sequences and a BsmBI restriction site (Supplementary Table 6)[57]. The PCR-amplified fragments were then ligated into the BsmBI-digested (New England Biolabs) SP199 in an equimolar ratio in a Golden Gate reaction using T4 ligase (New England Biolabs) and the BsmBI isoschizomer *Esp3I* (New England Biolabs) for 20 cycles (5 min at 37 °C and 5 min at 20 °C) with a final denaturation step at 65 °C for 20 min. Vectors were transformed into NEB Stable competent *Eschericia coli*. Successful assembly was verified by ApaI digest (New England Biolabs) and Sanger sequencing.

### Generation of reporter lines

The lentiviral FIREWACh system was used to create polyclonal GFP reporter lines of Xist-controlling RE, as described previously[33,58]. To this end, RE inserts were amplified from genomic DNA or bacterial artificial chromosomes (Supplementary Table 6). They were then inserted into the BamHI-digested (New England Biolabs) FpG5 vector (Addgene 69443) using the InFusion HD Cloning Kit (Takara)[58]. Vectors were transformed into NEB Stable competent *E. coli*. Successful cloning was verified by Sanger sequencing. The FIREWACh plasmids were integrated into the TX SP107 cell line using lentiviral transduction. Successful integration was verified using flow cytometry. A complete description of the generated lines is shown in Supplementary Table 6.

### Lentiviral transduction

CRISPR sgRNA expression vectors and FIRWACh reporter constructs were stably integrated into mES cells using lentiviral transduction. To this end, Lenti-X 293T cells were seeded at a density of $1.04 \times 10^5$ cells per cm² and transfected with a third generation transfer system consisting of VSVG, pLP1 and pLP2 (Thermo Fisher). For a six-well plate, 1.2 μg of PLP1, 0.6 μg of pLP2 and 0.4 μg of VSVG were incubated in 250 μl of OptiMEM (Thermo Fisher) with 2 μg of the transfer plasmid. After 5 min, 11.25 μl of Lipofectamine 2000 (Thermo Fisher), concomitantly incubated in 250 μl OptiMEM, was added. Following another incubation of 15 min, the mixture was transferred onto the Lenti-X cells. Viral supernatant was isolated after 48 h and concentrated 1:10 using the Lenti-X concentrator (Takara). Concentrated supernatant was resuspended in PBS and stored at −80 °C. Before lentiviral transduction, mES cells were cultured in SL conditions to prevent loss of the second X chromosome[12]. For transduction, cells were seeded at a density of $2.9 \times 10^4$ cells per cm². The next day, 8 ng μl⁻¹ polybrene (Merck) was added to the culture medium to enhance efficiency. A total of 50–200 μl of concentrated virus was used per transduction. Starting 48 h afterwards, successful integrations were enriched using antibiotic selection with puromycin (1 μg ml⁻¹; Sigma) or hygromycin B (0.2 mg ml⁻¹; Sigma or VWR). Unless indicated otherwise, cells were transferred to 2iSL medium and cultured for at least five passages before further experiments. XX status was assessed using RNA-FISH with probes targeting the introns of the X-linked gene *Huwe1* (Supplementary Table 6).

### Generation of TX CasTuner line

The TX SP427 cell line (clone B2), stably expressing the FKBP-dCas9-HDAC4 CasTuner system, was generated by *PiggyBac* transposition[29]. TX1072 A3 mES cells were reverse-transfected using Lipofectamine 3000 (Invitrogen) with the pSLPB2B-FKBP12_F36V-hHDAC4-SpdCas9-tagBFP-PGK-Blast plasmid (Addgene, 187956) and a vector carrying a hyperactive PiggyBac transposase (pBROAD3-hyPBase-IRES-zeocin) at a 5:1 molar ratio. Successful integrations were selected using blasticidin (5 μg μl⁻¹; Roth). The monoclonal B2 line was generated by fluorescence-activated cell sorting (FACS) based on tagBFP expression and next-generation sequencing (NGS) karyotyping to ensure a normal karyotype (40, XX).

### Real-time qPCR

Cells were lysed directly on the tissue culture plates using Trizol (Invitrogen). Next, RNA was extracted using the Direct-zol RNA purification

Kit (Zymo Research). Subsequently, RNA was reverse-transcribed into complementary DNA (cDNA) using Superscript III reverse transcriptase (Invitrogen) with random hexamer primers. Expression levels were quantified using the Quant-Studio 7 Flex real-time PCR machine (Thermo Fisher Scientific) with Power SYBR Green PCR master mix (Thermo Fisher). Primers used are listed in Supplementary Table 6.

### Single-molecule RNA-FISH

Exonic Xist RNA signals were quantified at the single-cell level using RNA-FISH. Hybridization was performed using Stellaris FISH probes (Biosearch Technologies) (Supplementary Table 6). To this end, cells were collected using Accutase (Invitrogen) and adsorbed onto coverslips (no. 1.5, 1 mm) coated with poly(L-lysine) (Sigma) for 5–10 min. The cells were then fixed with 3% paraformaldehyde in PBS for 10 min and permeabilized for 5 min in PBS containing 0.5% Triton X-100 (Sigma) and 2 mM ribonucleoside vanadyl complex (New England Biolabs). Coverslips were preserved for later use in 70% ethanol at −20 °C. Hybridization was carried out overnight at 37 °C with 250 nM FISH probe in 50 ml of Stellaris RNA-FISH buffer (Biosearch Technologies) containing 10% formamide. Coverslips were washed twice for 30 min at 37 °C, with 2× SSC (saline sodium citrate) and 10% formamide with 0.2 mg ml$^{-1}$ DAPI (Sigma) being added to the second wash. Before mounting with Vectashield (Biozol), coverslips were washed with 2× SSC at room temperature for 5 min. Images were acquired using a widefield Cell Discoverer 7 or Z1 Observer microscope (Zeiss) using a ×100 objective.

### Flow cytometry

Fluorescence activity of the FIREWACh reporter lines and cells stained by FlowFISH was assayed using flow cytometry[58]. For the reporter lines, cells were resuspended in FACS buffer (1% FBS in PBS with 1 mM EDTA; Thermo Fisher). Cells were analyzed and sorted using the FACSAria Fusion flow cytometer (BD Biosciences). At least 20,000 cells were assayed per measurement. Side and forward area scatters were used to gate for live cells. The height and width of the forward and sideward scatters were used to discriminate doublets.

### FlowFISH CRISPR screens

**CRISPR library cloning.** Both libraries were cloned into the BsmBI-digested SP199 expression vector[56]. The oligo pool of the TFi Lib was amplified with the primers OG113 and TS122 for 14 cycles in seven individual PCR reactions with the KAPA HiFi PCR ReadyMix (Roche). As the oligo pool of the TFiMini Lib was ordered with another library, it was amplified in two steps. First, the oligo pool was separated from the other library using the KAPA HiFi PCR ReadyMix with OG113 and LR256 for 12 cycles. Then, cloning overhangs were added using four separate PCR reactions with 500 ng each, using the primers OG113 and TS122. For both libraries, two Gibson cloning reactions were performed using 7 ng of the insert and 100 ng of SP199. The reactions were pooled, ethanol-precipitated and resuspended in 5 μl of water. The eluted DNA was transformed into 20 μl of MegaX DH10B electrocompetent cells (Thermo Fisher). Successful cloning was confirmed using Sanger sequencing and restriction digests with BsmBI and XhoI (New England Biolabs). The coverage of the libraries was determined as 513× (TFi Lib) and 8,472× (TFiMini Lib). Furthermore, both libraries were amplified using the KAPA HiFi PCR ReadyMix for 12 cycles to add sequencing overhangs. The TFi Lib was sequenced with 150-bp paired-end reads on the MiSeq platform, yielding ~1 × 10$^7$ fragments. The TFiMini Lib was sequenced with 100 bp paired-end reads on the NovaSeq 6000 platform, yielding ~4.3 × 10$^7$ fragments. A low log$_2$ distribution width of the guide counts of 2.2 (TFi Lib) and 1.5 (TFiMini Lib) confirmed that high coverage was retained during cloning (Extended Data Figs. 1h and 3f). All guides were present in both libraries.

**Titer estimation.** The cloned libraries were packaged into lentiviral particles as described above. Before transduction, the viral titers were estimated. To this end, the TX SP107 (TFi screen) and TX SP427 (TFiMini

screen) cell lines were transduced in six-well plates with tenfold serial dilutions of the respective lentiviruses (10$^{-2}$–10$^{-7}$) in duplicates. After 2 days, antibiotic selection was performed using puromycin (1 μg ml$^{-1}$; Sigma). After 7 days, the surviving colonies were counted in the wells. The viral titers were estimated as 6.8 × 10$^5$ transduction units (TU) per ml for the TFi screen and 9.4 × 10$^5$ TU per ml for the TFiMini screen.

**Tissue culture and cell sorting.** Both screens were performed in two replicates. A coverage of >300× was retained during all steps. For the TFi screen, cells transduced with nontargeting guides or guides targeting RE57 were cultured alongside the library as controls. Following transduction of the TFi Lib into the TX SP107 cell line under SL conditions (multiplicity of infection (MOI) = 0.3), the cells were selected for 3 days using puromycin. Subsequently, the cells were transferred into 2iSL medium. At the same time, 1 × 10$^7$ cells were frozen (selected population). After 3 days in 2iSL conditions the split dCas9–KRAB system was induced by the addition of ABA. The following day, the cells were differentiated for 48 h through 2i/L withdrawal and collected for FlowFISH staining.

For the TFiMini screen, the TX SP427 cell line was transduced under SL conditions (MOI = 0.3) in the presence of dTAG-13. Following 3 days of puromycin selection, the cells were transferred to 2iSL conditions (Extended Data Fig. 3b). At the same time, 1 × 10$^7$ cells were frozen (Selected population). After 8 days, the CasTuner system was induced by the removal of dTAG-13 from the medium. A flask with medium containing dTAG-13 was taken along as a control. Then, 2 days later, the cells were differentiated through 2i/LIF withdrawal. At the same time, 1 × 10$^7$ cells were frozen (2iSL population) to confirm the inducibility of the system. After 2 days of 2i/L withdrawal, the cells were collected for FlowFISH staining.

**FlowFISH.** The PrimeFlow RNA Assay Kit (Thermo Fisher) was used to assay Xist RNA. The assay was performed in conical 96-well plates with 5 × 10$^6$ cells per well. Xist RNA was labeled with Alexa-Fluor647 using a type 1 PrimeFlow probe (VB1-14258, Thermo Fisher). Lastly, cells were resuspended in PrimeFlow RNA storage buffer (Themo Fisher) and analyzed using flow cytometry. For the TFi screen, 1 × 10$^7$ cells were frozen during the protocol (Unsorted population). Cells were sorted according to Xist expression into Xist$^{High}$ (top 15%), Xist$^{Lowl}$ (bottom 15% of the Xist$^+$ cells) and Xist$^{Neg}$ (bottom 15%) populations. Xist$^+$ cells were determined on the basis of the 99th percentile of the Xist signal from a 2iSL sample (cells that do not express Xist). At least 1.5 × 10$^7$ cells were recovered per population. For the TFiMini screen, only Xist$^{High}$ and Xist$^{Neg}$ populations were sorted. At least 5 × 10$^5$ cells were recovered per population.

**DNA isolation and library preparation.** Sequencing libraries were prepared from all indicated populations. To this end, genomic DNA was isolated using phenol–chloroform extraction. First, cell pellets were incubated for 14 h at 65 °C in 250 μl of decrosslinking buffer (1% SDS (Invitrogen), 1.25 μl of DTT (Roth) and 10 μl of 5 M NaCl (Sigma) in Tris–EDTA buffer (Sigma)). Next, 20 μl of RNAse A (10 mg ml$^{-1}$; New England Biolabs) was added and the solution was incubated for 1 h at 37 °C. Next, 5 μl of proteinase K (20 mg ml$^{-1}$; Ambion) was added and the solution was incubated for 1 h at 50 °C. Subsequently, 275 μl of phenol–chloroform (Roth) was added and the mixture was vortexed for 1 min. The samples were then centrifuged for 10 min at 12,100$g$. The aqueous phase, containing the genomic DNA, was transferred to a new tube and the sample was cleaned using ethanol precipitation. The pellets were dried and resuspended in 50 μl of water.

The guide cassette was amplified for 20 cycles using the KAPA HiFi PCR ReadyMix with primers OG115 and OG116. To keep 300× coverage, at least 20 μg (TFi screen) or 1.6 μg (TFiMini screen) of genomic DNA was amplified per sample. Between 0.1 and 2 μg of genomic DNA was amplified per reaction, as the PCR tended to be inhibited in samples

stained using the FlowFISH protocol[12]. Subsequently, PCR reactions of the same samples were pooled and concentrated using the DNA clean and concentrator kit (Zymo Research). Lastly, sequencing barcodes were added in a second PCR using the KAPA HiFi PCR ReadyMix for 11–12 cycles (primer sequences in Supplementary Table 6). The TFi screen was sequenced with 100-bp paired-end reads on the NextSeq 500 platform, yielding ~2–7 × 10⁶ fragments per sample. The TFiMini screen was sequenced 100-bp paired-end reads on the NextSeq 2000 platform, yielding ~4–8 × 10⁶ fragments per sample.

## Reporter screens

The viral titer of the TFiMini library for the reporter screens (2.6 × 10⁶ TU per ml) was estimated in the TX SP107 cell line under 2iSL conditions, as described for the FlowFISH screens. The screens were performed in three replicates with >200× coverage. The TFiMini library was transduced under 2iSL conditions into the TX SP107 cell line carrying a polyclonal insertion of the FIREWACh reporter construct (Supplementary Table 6) with one of 12 RE inserts (noRE, RE57L, RE57M, RE57R, RE58, RE61, RE85, RE93, RE95, RE96, RE97 or RE127)[58]. Following 3 days of puromycin selection, ABA was added to the medium to induce the split dCas9–KRAB system. Differentiation was induced 24 h later using 2i/LIF withdrawal. Before sorting, 2 × 10⁶ cells were taken per sample as the unsorted population. Flow cytometry was used to sort cells according to their GFP expression into GFP^high (top 10%) and GFP^low (bottom 10%) populations. At least 7 × 10⁵ cells were recovered per population. Genomic DNA was isolated as described above for the FlowFISH screens. The guide cassette was amplified for 20–25 cycles using the KAPA HiFi PCR ReadyMix with primers OG115 and OG116. At least 2.2 μg of genomic DNA was amplified per sample. Following cleanup with the DNA clean and concentrator kit, sequencing barcodes were added in a second PCR using the KAPA HiFi PCR ReadyMix for 9–11 cycles (primers in Supplementary Table 6). The samples was sequenced with 100-bp paired-end reads on the NovaSeq 6000 platform, yielding ~0.2–8 × 10⁶ fragments per sample.

## Poly(A)-enriched RNA-seq

Poly(A)-enriched RNA-seq was performed using the Collibri 3′ mRNA library prep kit (Thermo Fisher) in TX1072 XX A3 and XO B7 cells at 0, 10, 16, 24, 30, 36, 48, 56, 72 and 96 h of 2i/LIF withdrawal in three biological replicates. Library preparation was performed according to manufacturer's instructions using 500 ng of RNA per sample. The amplified samples were sequenced with 100-bp paired-end reads using the NovaSeq 6000, yielding ~1.6–16 × 10⁶ fragments per sample.

## scRNA-seq

scRNA-seq was performed in two replicates on day 4 of 2i/LIF withdrawal in a homozygous ΔFtx–Xert deletion line[12], TX1072 XX A3 wild-type control and the sgXist-Promoter/sgXist-Distal REs CasTuner lines at six different dTAG concentrations, using the Single-Cell 3′ reagent kit v3.1 (10X Genomics). The sequencing libraries of the ΔFtx–Xert deletion experiment were prepared together with several samples of an unrelated study.

For sample multiplexing, MULTI-seq barcode–lipid complexes were used according to the published protocol with minor modifications[59]. Sample barcodes were chosen from a list of compatible barcodes[60]. Lipid-modified anchor and coanchor oligos were purchased from Sigma-Aldrich. Barcode and library preparation oligos were purchased from Eurofins with NGS-grade quality. Per sample, 1 × 10⁵ cells were transferred to a 96-well ultralow-attachment plate (Costar) and kept on ice during the procedure. After two washes with PBS, the cells were incubated for 5–10 min with an anchor–barcode solution (45 μl of PBS, 5 μl of anchor–barcode). The mixture was incubated for another 5–10 min after adding 5 μl of coanchor solution. Labeling was quenched by adding 1% BSA in PBS; samples were washed once in the same solution and resuspended in 0.4% BSA in PBS. Afterward, cells were counted, pooled equally and filtered using a Flowmi cell strainer 40 μm (Sigma-Aldrich). A total of 2.5 × 10⁴ cells were used for scRNA-seq gene

expression library preparation. The 10x Genomics library preparation protocol was performed according to the manufacturer's instructions (CG000315 Rev E, 10x Genomics) with the additional steps required to generate the separate MULTI-seq libraries for the sample-to-cell barcode assignment[59]. During cDNA amplification, 0.5 μl of 1.25 μM 'MULTI-seq additive primer' was added. Then, 50% of the purified gene expression cDNA was carried forward and the cycle number in the index PCR was reduced by one accordingly to increase library complexity[61]. During the first step of cDNA cleanup, the supernatant was kept, which contained the short MULTI-seq library. This MULTI-seq library was then cleaned up two times using solid-phase reversible immobilization (SPRI) beads (KAPA HyperPure beads, Roche). Sequencing adaptors containing indices ('TruSeq_RPIX_' and 'Universal_I5_with_index') were added by PCR, using 2.5 μl of primer at 10 μM each, 10 ng of cDNA, 26.25 μl of Kapa HiFi HotStart ReadyMix 2× and water to a 50-μl final volume, with a program of 95 °C for 5 min and 11 cycles of 98 °C for 15 s, 60 °C for 30 s, 72 °C for 30 s, followed by 72 °C for 1 min and a 4 °C hold. The resulting sequencing library was then cleaned up another time with SPRI beads. For the ΔFtx–Xert deletion experiment, 10x gene expression libraries and MULTI-seq libraries were sequenced with asynchronous 90-bp and 28-bp paired-end reads on the NovaSeq 6000 platform, yielding a minimum of ~6.2 × 10⁸ and ~4.4 × 10⁷ fragments, respectively. For the Xist titration experiment, 10x gene expression libraries and MULTI-seq libraries were sequenced with asynchronous 100-bp and 28-bp paired-end reads on the Aviti platform (Element Biosciences), yielding a minimum of ~8.2 × 10⁸ and ~4.8 × 10⁷ fragments, respectively.

## CUT&Tag of histone modifications

CUT&Tag was performed in three biological replicates to map active histone modifications along the genome, as described previously[62]. The assay was conducted for H3K27ac, H3K4me3 and H3K4me1 on day 2 of 2i/LIF withdrawal in TX SP427 mES cells transduced with guides targeting different Xist activators (sgZic3, sgNfrkb, sgOtx2 and sgFoxd3) or a nontargeting control (sgNT). Following dissociation with Accutase, 1 × 10⁵ cells per antibody were collected and quickly washed in wash buffer (20 mM HEPES–KOH pH 7.5, 150 mM NaCl, 0.5 mM spermidine, 10 mM sodium butyrate, protease Inhibitor and 1 mM PMSF). Then, 10 μl of concanavalin A beads (Bangs Laboratories) were equilibrated with 100 μl of binding buffer (20 mM HEPES–KOH pH 7.5, 10 mM KCl, 1 mM CaCl₂ and 1 mM MnCl₂) and then concentrated in 10 μl of binding buffer. The cells were bound to the concanavalin A beads by incubating for 10 min at room temperature on a rotator. Next, the beads were separated using a magnet and resuspended in 100 μl of chilled antibody buffer (wash buffer with 0.05% digitonin and 2 mM EDTA). Subsequently, 1 μl of primary antibody (Supplementary Table 6) was added and incubated on a rotator for 3 h at 4 °C. After magnetic separation, the beads were resuspended in 100 μl of chilled Dig-wash buffer (wash buffer with 0.05% digitonin) containing 1 μl of secondary antibody and incubated for 1 h at 4 °C on a rotator. The beads were washed three times with chilled Dig-wash buffer and resuspended in chilled Dig-300 buffer (20 mM HEPES–KOH pH 7.5, 300 mM NaCl, 0.5 mM spermidine, 0.01% digitonin, 10 mM sodium butyrate and 1 mM PMSF) with 1:250 diluted 3×FLAG–pA-Tn5 preloaded with mosaic-end adaptors. After incubation for 1 h at 4 °C on a rotator, the beads were washed four times with chilled Dig-300 buffer and resuspended in 50 μl of tagmentation buffer (Dig-300 buffer and 10 mM MgCl₂). Tagmentation was performed for 1 h at 37 °C and stopped by adding 2.25 μl of 0.5 M EDTA, 2.75 μl of 10% SDS and 0.5 μl of 20 mg ml⁻¹ proteinase K and vortexing for 5 s. DNA fragments were solubilized overnight at 55 °C followed by 30 min at 70 °C to inactivate residual proteinase K. DNA fragments were purified with the ChIP DNA clean and concentrator kit (Zymo Research) and eluted with 25 μl of elution buffer.

Sequencing libraries were generated by amplifying the DNA fragments with barcoded primers using NEBNext HiFi 2× PCR master mix for 14 cycles (Supplementary Table 6). Cleanup following PCR was performed with a 1× volume of Ampure XP beads (Beckman Coulter)

and samples were eluted in 27 μl of 10 mM Tris pH 8.0. The samples were sequenced with 100-bp paired-end reads on the NovaSeq 6000 platform, yielding ~4.8–7.9 × 10⁶ fragments per sample. Information on antibodies is supplied in Supplementary Table 6.

### Statistics and reproducibility

Statistical analysis was conducted in RStudio (version 4.2). No statistical methods were conducted to determine sample size. For the bulk RNA-seq time course, one replicate sample (X0_36h_RI) was excluded because of low read number. The experiments were not randomized. The investigators were not blinded to allocation during experiments and outcome assessment.

### Computational methods

Computational methods are described in Supplementary Note 3.

### Reporting summary

Further information on research design is available in the Nature Portfolio Reporting Summary linked to this article.

### Data availability

Sequencing data generated in this study are available from the Gene Expression Omnibus under accession number GSE274507. RNA-FISH microscope images are available from Zenodo: https://doi.org/10.5281/zenodo.12821363 (ref. 63) (related to Fig. 2, *Oct4* knockdown), https://doi.org/10.5281/zenodo.12821095 (ref. 64) (related to Fig. 5, Xist activator knockdown) and https://doi.org/10.5281/zenodo.15617117 (ref. 65) (related to Extended Data Fig. 10, Xist-RE knockdown). Flow cytometry data are also available from Zenodo (https://doi.org/10.5281/zenodo.12822424)⁶⁶ (related to Figs. 1 and 4). AnimalTFDB 3.0, used to design the TF screen library, is available online (https://guolab.wchscu.cn/AnimalTFDB/#!/). The mm10 reference genome used in this study is also available online (https://hgdownload.soe.ucsc.edu/goldenPath/mm10/bigZips/latest/mm10.fa.gz). Source data are provided with this paper.

### Code availability

Code used in this study is available from GitHub (https://github.com/EddaSchulz/TFiScreen_Paper).

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

### Acknowledgements

We thank the FACS facility, microscope facility, sequencing facility and IT service at the Max Planck Institute for Molecular Genetics for support. We thank J. Tünnermann and R. Buschow for help with setting up the automated FISH analysis and A. Bolondi for advice on MULTI-seq sample labeling. This work was supported by the Max Planck Lise Meitner Excellence program, European Research Council Starting Grant CisTune (948771) and German Research Foundation (DFG) priority program SPP 2202 (508000619) to E.G.S. T.S. and A.A. were supported by the DFG (IRTG 2403). G.N. was supported by the European Union's Horizon 2020 Research and Innovation Program (Marie Skłodowska-Curie ITN PEP-NET). The funders had no role in study design, data collection and analysis, decision to publish or preparation of the manuscript.

### Author contributions

T.S. and E.G.S. conceptualized the project. G.N. generated the TX1072 XX SP427 cell line and performed RNA-seq. E.K. performed CUT&Tag. J.J.F. performed the scRNA-seq. M.B. and J.J.F. processed scRNA-seq data. A.M. performed validation experiments. A.M. and H.M. generated the sgXist-Distal-REs CRISPRi line. A.A. integrated in vitro RNA-seq with in vivo scRNA-seq data. J.S. performed EpiLC differentiations. T.S. and V.-Y.F. generated FIREWACh reporter plasmids and performed reporter time-course experiments. T.S. performed all other experiments and analyses. T.S. and E.G.S. wrote the paper. E.G.S. and M.V. acquired funding.

### Funding

### Competing interests

The authors declare no competing interests.

### Additional information

**Extended data** is available for this paper at https://doi.org/10.1038/s41594-025-01686-3.

**Correspondence and requests for materials** should be addressed to Edda G. Schulz.

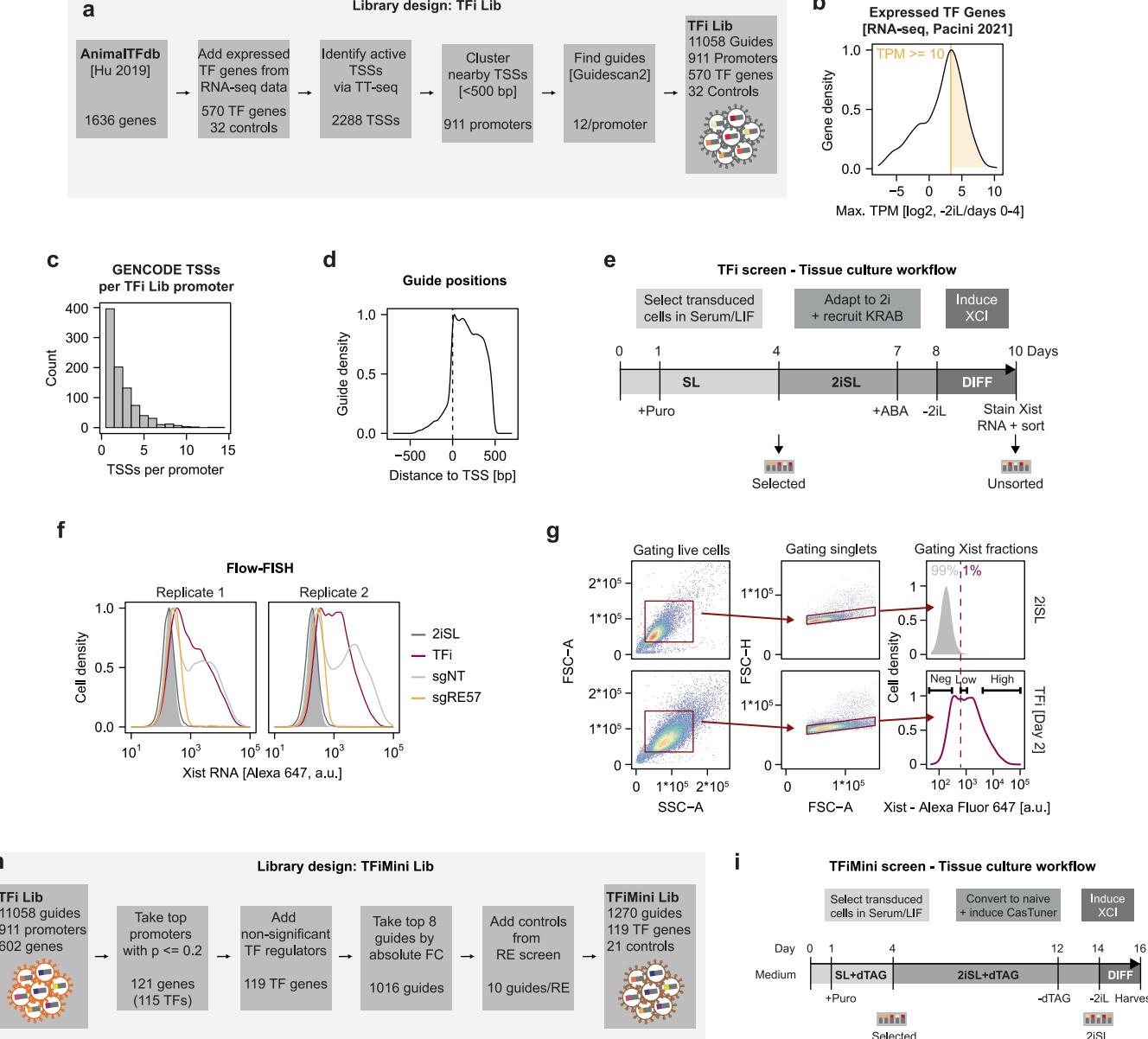

**Extended Data Fig. 1 | CRISPR screen design.** (**a**) Workflow to design the TFi library. Expressed TF genes during the first 4 days of 2iL-withdrawal were filtered from the *AnimalTFdb 3.0* using a published RNA-seq time course (maximal TPM > 10). Active TSSs were then detected using published TT-seq data. Nearby TSSs (≤500 bp) of the same genes were grouped into promoters. *Guidescan2* was used to generate up to 12 guides per promoter. (**b**) Selection of expressed TF genes using a published RNA-seq time course of 2iL-withdrawal. Max TPM denotes the highest expression value between days 0-4 of differentiation. The chosen threshold of 10 TPM is indicated by the orange line. (**c**) Histogram depicting the number of annotated TSSs per combined promoter. (**d**) Density plot showing the location of the designed guides relative to annotated TSSs. (**e**) Experimental workflow for the TFi screen. Icons below the timeline indicate the point at which cells were harvested for the Selected and Unsorted populations. (**f**) Xist RNA expression assessed by flow cytometry following

FlowFISH staining. Cells transduced with the library (TFi), with a non-targeting (sgNT) and a positive control (sgRE57) guide, and an undifferentiated sample (2iSL) are shown. (**g**) Example plot describing the gating strategy during the TFi screen. The Xist[Low] population was gated based on the Xist levels of the 99th percentile of the 2iSL sample (Xist- cells). (**h**) Workflow of the design of the TFiMini library. The promoters outside of the *Xic*, included in the TFi library, were filtered based on their enrichment in the Xist[High]/Xist[Neg] comparison (Wald.p ≤ 0.2). Non-enriched published TF regulators were added, despite missing the significance threshold. The top 8 guides were selected by absolute fold change in the Xist[High]/Xist[Neg] comparison in the TFi screen. Lastly, guides targeting Xist-controlling REs and the Fgf4 minimal promoter, used in the reporter screens, were added as controls. (**i**) Experimental workflow for the TFiMini screen. Icons below the timeline indicate the point at which cells were harvested for the Selected and 2iSL populations.

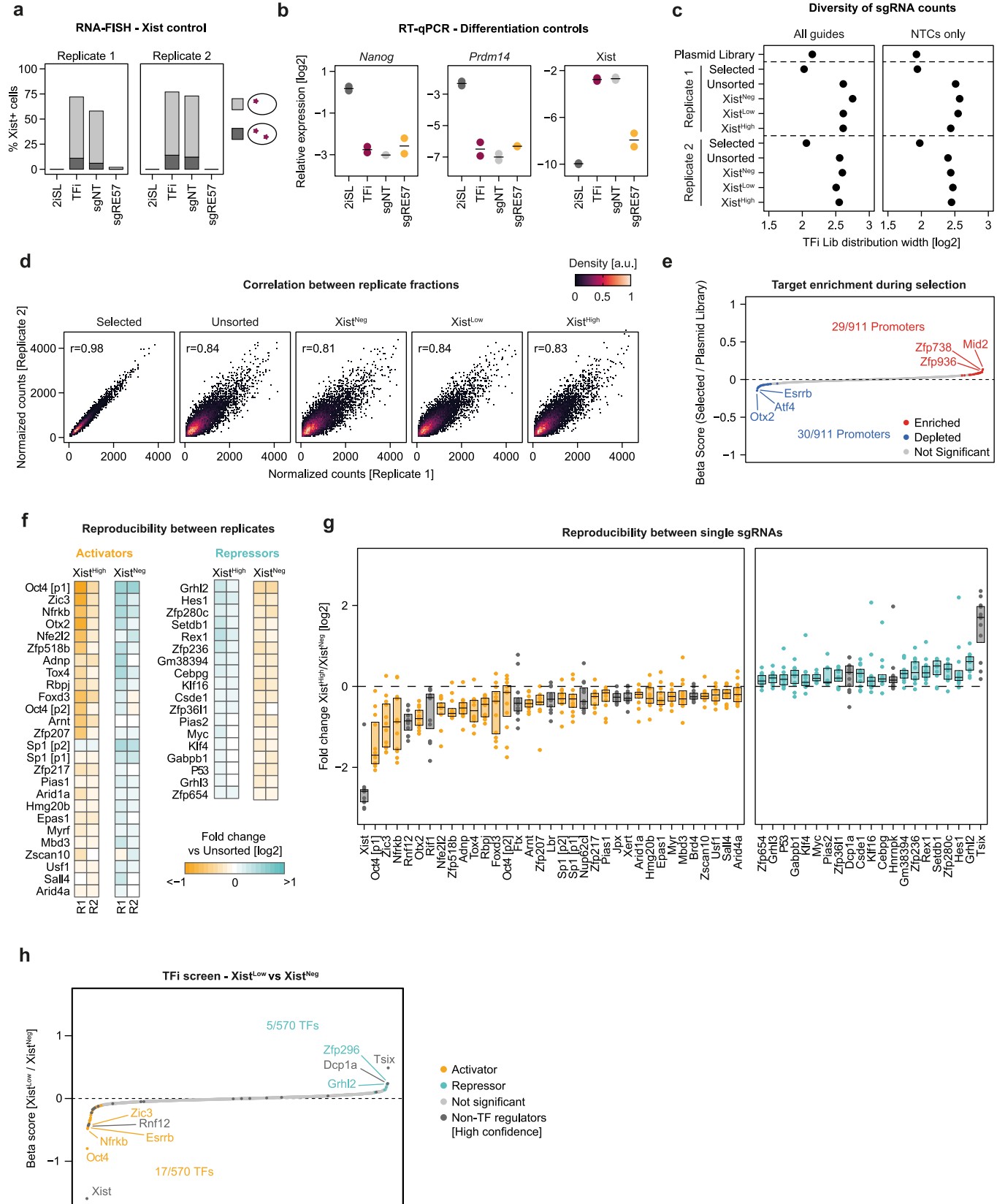

**Extended Data Fig. 2 | See next page for caption.**

**Extended Data Fig. 2 | Quality controls and screening results of the TFi CRISPR screen.** (**a**) Percentage of Xist+ cells assessed using RNA-FISH. The fraction of cells with biallelic Xist signals is indicated in dark gray. (**b**) Expression of Xist and the naive pluripotency factors *Prdm14* and *Nanog* assessed by RT-qPCR. The black bars denote the mean between two replicates (dots). (**c**) sgRNA distribution width of the normalized counts in the different populations of the TFi screen, grouped into targeting (left) and non-targeting guides (right). The score is given by the fold change between the 90th and the 10th percentile. (**d**) Correlation between the two replicates in the different populations of the TFi screen (11,058 guides). The Pearson correlation coefficient is indicated in the plots. (**e**) Ranked plot depicting enriched and depleted targets between the plasmid library and the Selected population. Significance was assessed using *MAGeCK mle* (Wald.p ≤ 0.05). (**f**) Mean log$_2$ fold change across guides of the

Xist$^{High}$ and Xist$^{Neg}$ populations compared to the Unsorted cells, separated by replicates (R1/R2). All significant activators and repressors from the Xist$^{High}$/Xist$^{Neg}$ comparison were included in the plot (Wald.p ≤ 0.05). (**g**) Log$_2$ fold change of the individual guides between the Xist$^{High}$ and Xist$^{Neg}$ populations in the TFi screen (12 guides per targeted promoter). TF activators and repressors are colored in orange and teal, respectively. Non-TF controls are colored in gray. The boxes indicate the interquartile range (IQR, 25th to 75th percentiles), as well as the median (black line). (**h**) Ranked plot depicting enriched and depleted targets between Xist$^{Low}$ and Xist$^{Neg}$ populations in the TFi screen. Significantly enriched or depleted targets are colored in teal or orange respectively (*MAGeCK mle*, Wald.p ≤ 0.05). High confidence non-TF controls are colored in dark gray (see Supplementary Table 1).

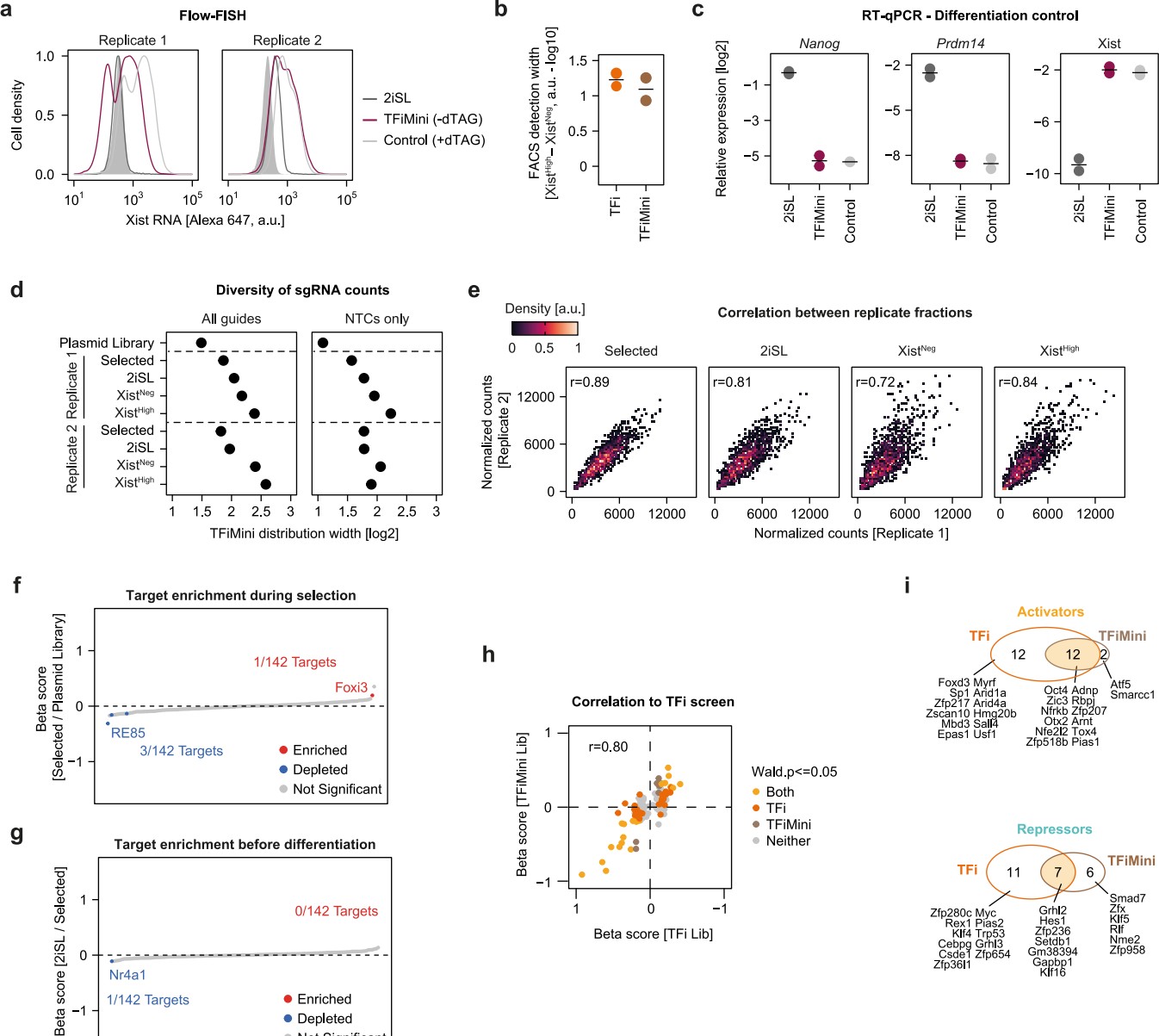

**Extended Data Fig. 3 | Quality controls and screening results of the TFi Mini CRISPR screen. (a)** Xist RNA expression assessed by flow cytometry following FlowFISH staining. Additionally to the library (TFiMini) the plot also depicts an undifferentiated (2iSL) and a +dTAG control (CRISPR system not induced). **(b)** Comparison of the Xist detection range across screens and replicates. The detection range was quantified as the difference between the log$_{10}$-transformed MFI values of the Xist$^{HIgh}$ and Xist$^{Neg}$ gates. **(c)** Expression of Xist and the naive pluripotency factors *Prdm14* and *Nanog* during the TFiMini screen assessed by RT-qPCR. The black bars denote the mean between two replicates (dots). **(d)** The sgRNA distribution width of the normalized counts in the different populations of the TFiMini screen, separated by targeting (left) and non-targeting guides (right). The score is given by the subtraction of the log$_2$-transformed values for the 90$^{th}$ and 10$^{th}$ percentile. **(e)** Correlation between the two replicates in the different populations of the TFiMini screen (1,270 guides). The Pearson correlation coefficient is indicated in the plots. **(f)** Ranked plot depicting enriched and depleted targets between the Selected population and the plasmid library in the TFiMini screen. Significance was assessed using *MAGeCK mle* (Wald.p ≤ 0.05). **(g)** Ranked plot depicting enriched and depleted targets between the 2iSL and Selected populations in the TFiMini screen. Significance was assessed using *MAGeCK mle* (Wald.p ≤ 0.05). **(h)** Correlation between the Xist$^{High}$/Xist$^{Neg}$ comparisons of the TFi and TFiMini screens (119 shared TF genes). Significantly enriched targets (Wald.p ≤ 0.05) are colored in light orange (both screens), dark orange (TFi screen only) or brown (TFiMini screen only). The Pearson correlation coefficient is indicated in the plot. **(i)** Venn diagram showing the overlap in significant activators (top) and repressors (bottom) (Wald.p ≤ 0.05) between the Xist$^{High}$/Xist$^{Neg}$ comparisons of the TFi and TFiMini screens.

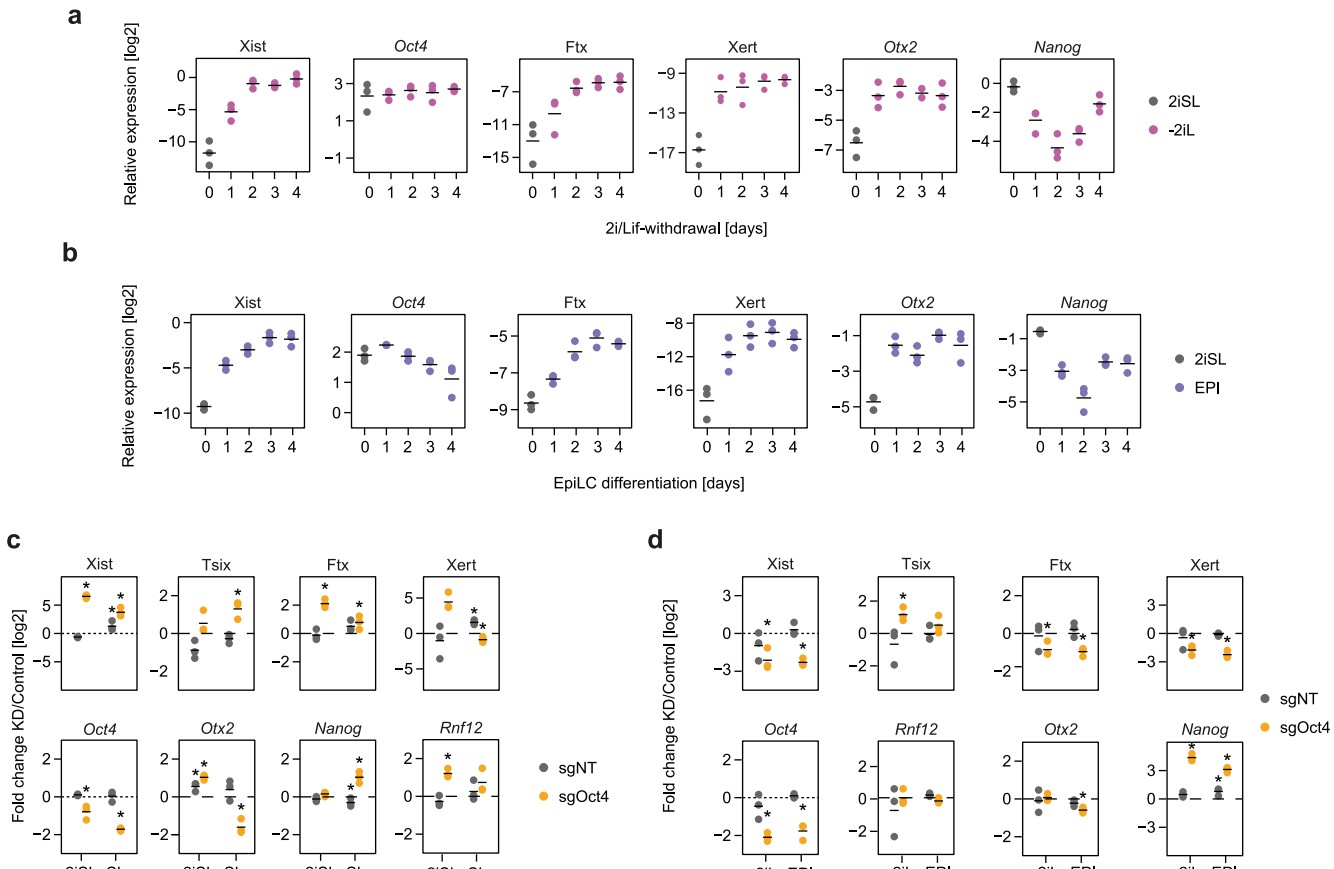

**Extended Data Fig. 4 | Oct4 depletion can activate or repress Xist depending on cellular context.** (**a-b**) RT-qPCR results for Xist, *Oct4* and other selected genes during 2iL-withdrawal (**a**) and during differentiation toward EpiLCs (**b**). The black bars denote the means across three replicates. (**c**) Effect of *Oct4* knockdown on the expression of selected Xist regulators in undifferentiated conditions

(2iSL or SL), assessed via RT-qPCR. The results are shown as a log$_2$ fold change between the control and knock down conditions. Significant differences are marked with a black asterisk (*two-sided unpaired t-test*, p ≤ 0.05). The black bars denote the means across three replicates. (**d**) As (**c**), but after two days of differentiation via 2iL-withdrawal (−2iL) or towards EpiLCs (EPI).

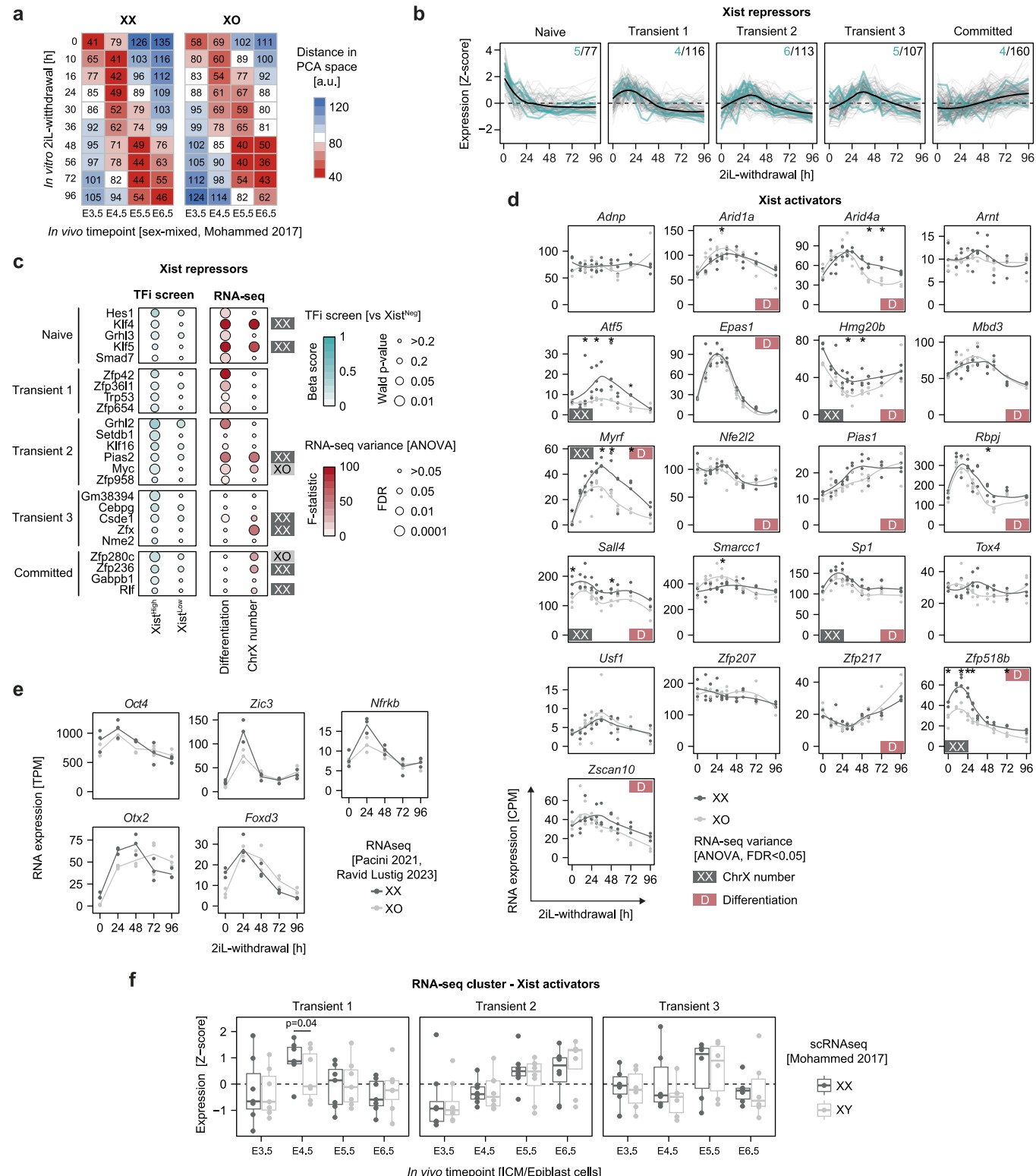

**Extended Data Fig. 5 | See next page for caption.**

**Extended Data Fig. 5 | Characterization of identified Xist repressors and activators.** (**a**) Pairwise distance in PCA space between published scRNA-seq data of developing mouse embryos and the XX or XO 2iL-withdrawal RNA-seq time course. Sex-mixed cells from the *in vivo* samples were filtered for ICM and epiblast cells and aggregated as pseudo-bulk. (**b**) Clustering of z-score transformed XX expression dynamics for all TF genes included in the TFi screen (see Supplementary Note 3). Xist repressors are shown in teal. The total numbers of genes and repressors per cluster is indicated in the top right. (**c**) Dotplot summarizing the TFi screen results and RNA-seq analysis for all Xist repressors identified in the TFi and TFi Mini screens. Statistics (MAGeCK mle, Wald.p-value) and effect size (Beta score) for the Xist$^{High}$/Xist$^{Neg}$ and Xist$^{Low}$/Xist$^{Neg}$ comparisons are shown on the left (blue). Contribution of differentiation or X-chromosome number to the expression variance in the RNA-seq time course (two-way ANOVA)

are shown on the right (red). The directionality for the X-chromosome effect is indicated next to the plot (FDR ≤ 0.05). (**d**) RNA expression dynamics of identified Xist activators. Difference between XX and XO cells was assessed per time point using *DESeq2* (Wald.FDR ≤ 0.05). Results from the ANOVA analysis are indicated as D (affected by differentiation) or XX (affected by X-chromosome number). (**e**) RNA expression dynamics of selected Xist activators in a published 2iL-withdrawal time course. (**f**) Z-score transformed expression of the transient Xist activator expression clusters during peri-implantation mouse embryo development. Expression was quantified in pseudobulk, separated by time point and sex. The dots represent individual genes (n = 7 for Transient 1, n = 6 for Transient 2/3). The boxes indicate the IQR, as well as the median (black line). Whiskers extend to the furthest values within 1.5 x IQR from the box. Significance was assessed using a *two-sided paired t-test* (p < 0.05).

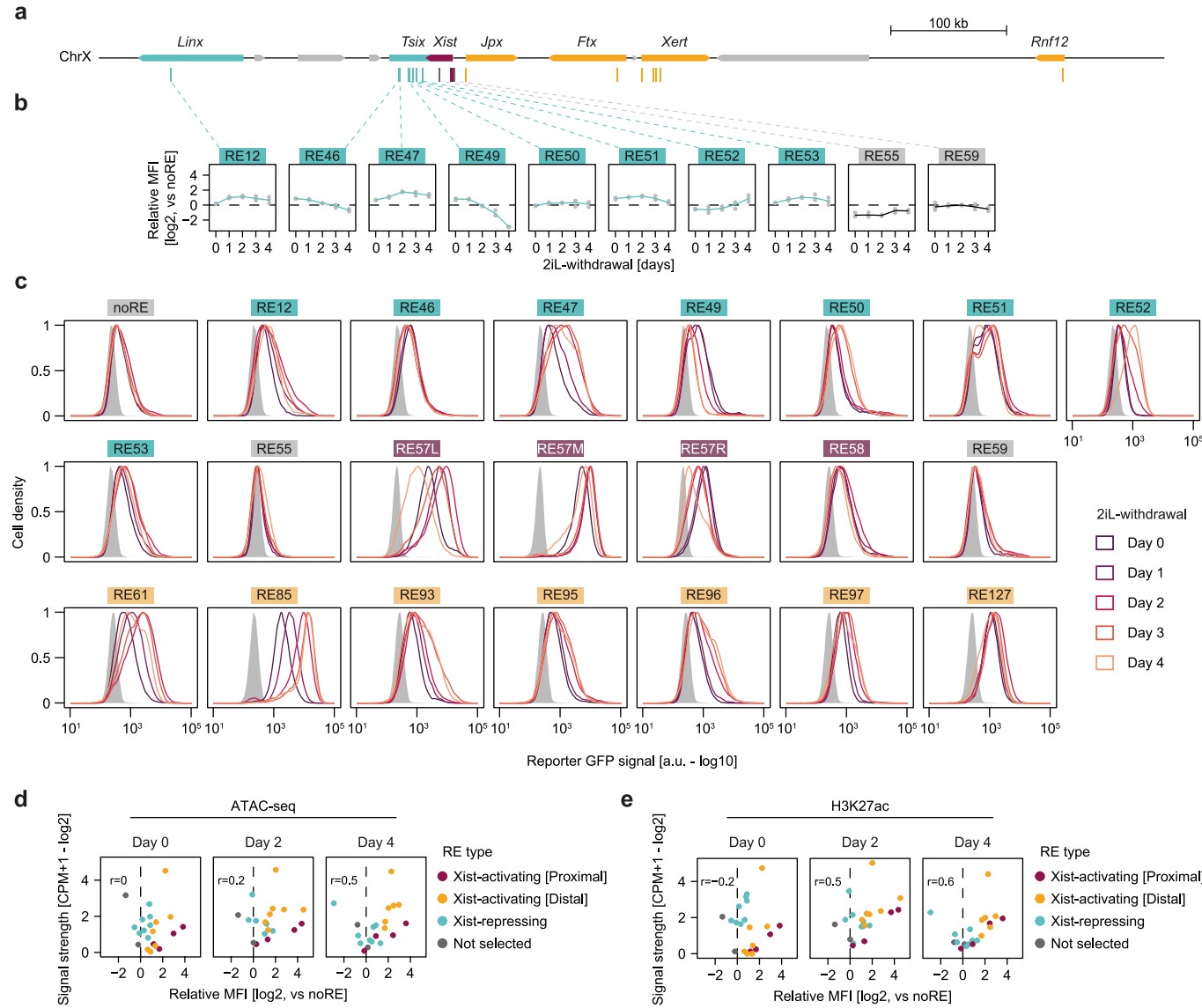

**Extended Data Fig. 6 | Characterization of reporter cell lines for Xist-controlling REs.** (**a**) Schematic outline the genomic region surrounding the *Xist* locus. Xist regulators are shown in teal (repressive) or orange (activating). Xist-controlling REs included in the reporter time course are shown in teal (Xist-repressive), gray (not selected for reporter screens), burgundy (proximal) or orange (distal). (**b**) Reporter expression dynamics for Xist-repressive (teal) and non-selected REs (gray). Relative MFI was calculated as the log$_2$ fold change between GFP levels in the indicated reporter and the noRE control. Mean (line) of n = 3 biological replicates (dots) is shown. (**c**) Reporter GFP signal assayed by

flow cytometry for the data summarized in (**b**) and Fig. 4c. One representative replicate is shown. The gray shade corresponds to non-fluorescent control cells. (**d**) Correlation between the reporter time course (mean of 3 biological replicates) and published chromatin accessibility data (mean of 2 biological replicates). The ATAC-seq data was generated in a female heterozygous TX Δ*Xic* deletion line. The assayed counts thus correspond to the Xi. The Pearson correlation coefficient is indicated in the plots. (**e**) As (**d**), but using published CUT&Tag data targeting H3K27ac.

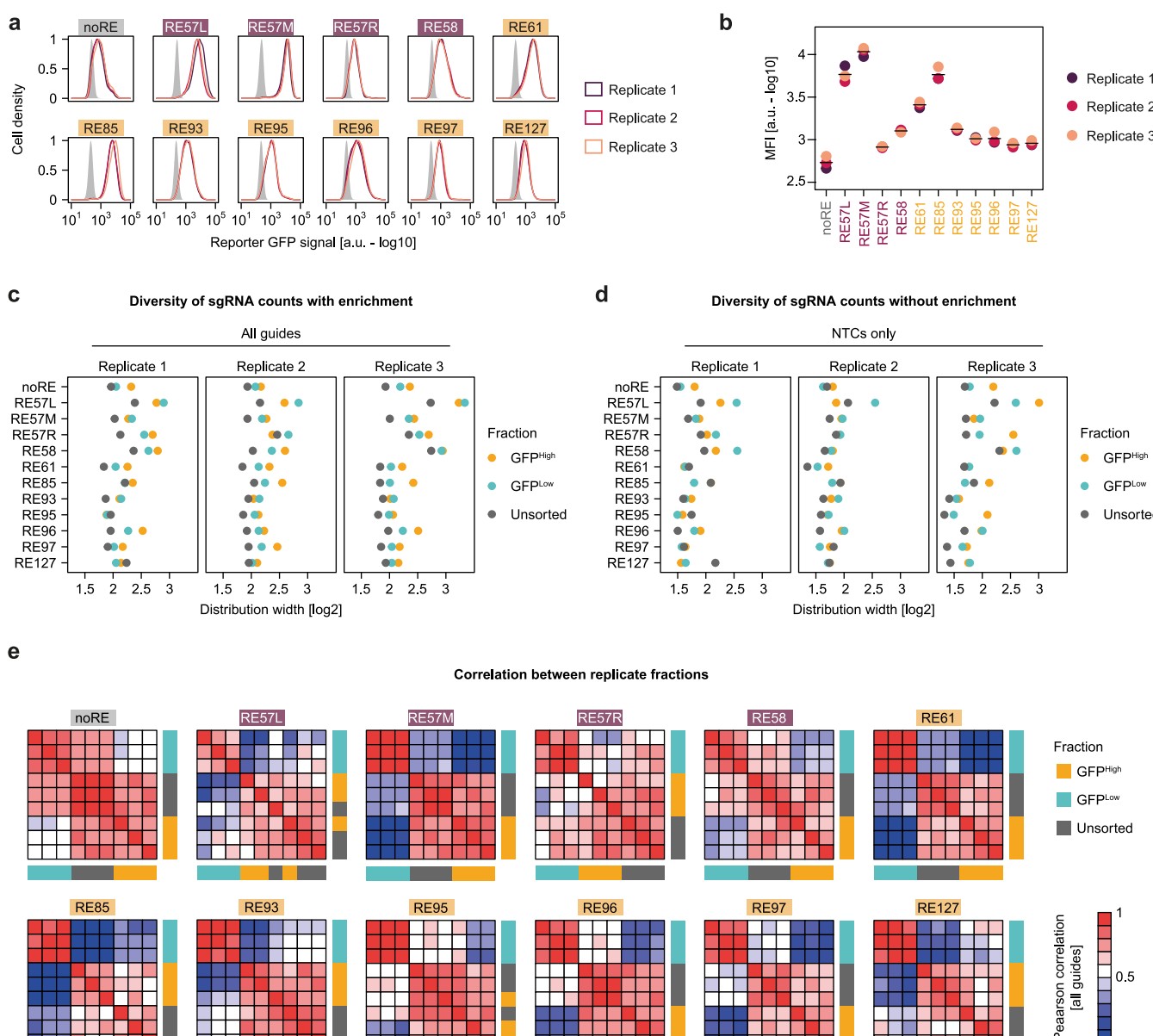

**Extended Data Fig. 7 | Quality controls for reporter screens.** (**a**) Reporter GFP signal during the reporter screens. The gray shade corresponds to non-fluorescent control cells. (**b**) Quantification of the data depicted in (**a**) as the mean fluorescence intensity. The black lines denote the geometric mean across three replicates. (**c**) SgRNA distribution width of the normalized counts in the different populations of the reporter screens, including all targeting guides. The score is given by the subtraction of the log$_2$-transformed values for the 90th and 10th percentiles. (**d**) As in (**a**), but including only non-targeting controls. (**e**) Correlation of the normalized counts between all replicate samples per reporter screen. The populations are indicated by colored boxes. The axes are ordered by hierarchical clustering.

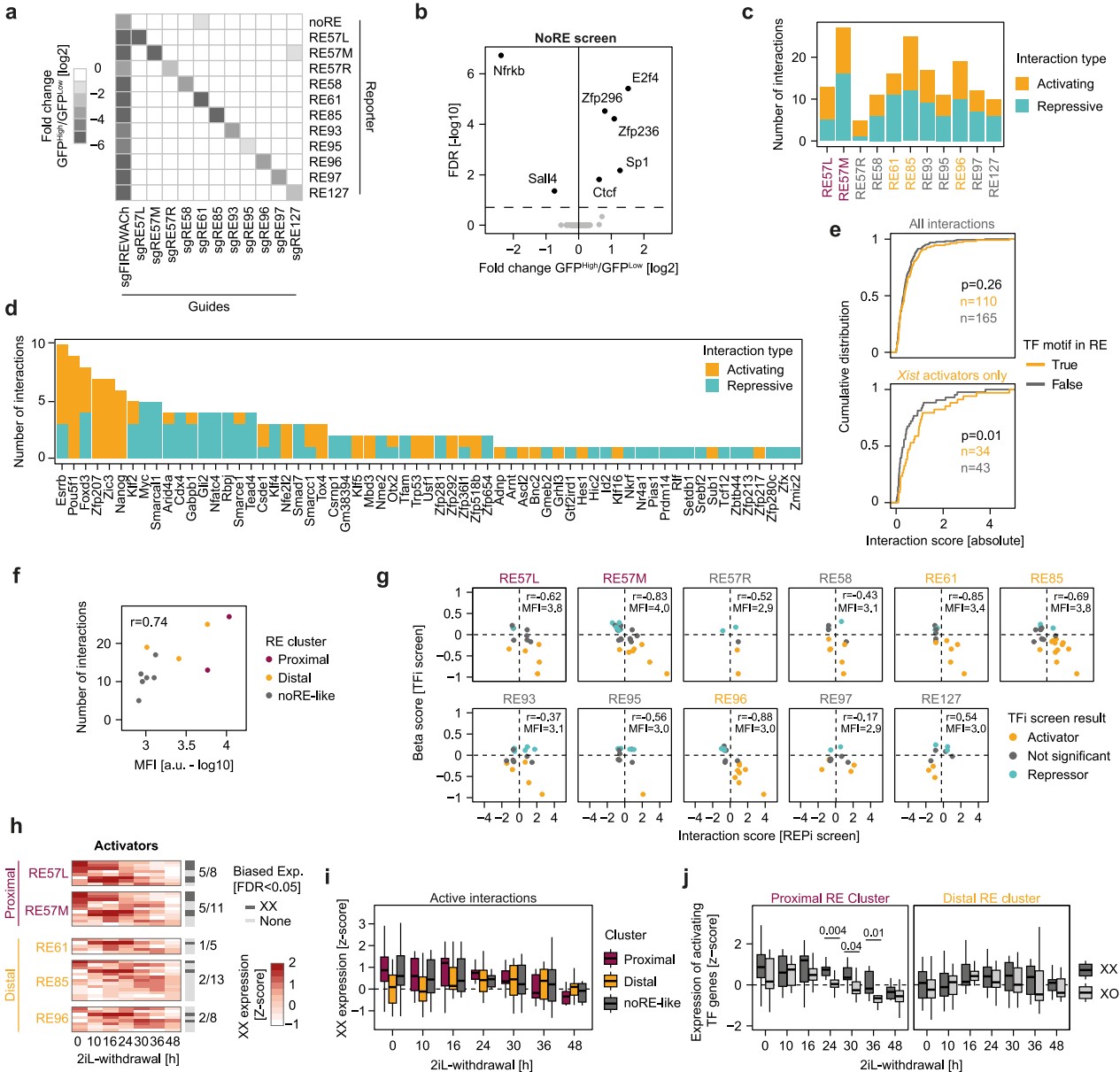

**Extended Data Fig. 8 | Reporter screen results.** (**a**) Effects of positive controls in the reporter screens. Fold change of the normalized counts between the GFP^High and GFP^Low populations for guides targeting the FIREWACh minimal promoter or the respective RE sequences. The heatmap denotes the mean across 3 biological replicates. (**b**) Enrichment of TF targets in the noRE screen. All 7 significant hits (labeled) were removed from further analysis (*two-sided unpaired t-test*, FDR ≤ 0.2). (**c-d**) Number of activating and repressive interactions per reporter line (**c**) and per TF target (**d**). (**e**) Cumulative distribution of the absolute interaction scores showing the correlation between the presence of a binding motif and interaction strength. The bottom panel shows only interactions of Xist activators from the TFi screen (Xist^High/Xist^Neg comparison). The results of a *rank-sum Wilcoxon test* and the number of tested interactions with (orange) and without (gray) a motif is indicated in the panels. (**f**) Correlation between base signal strength of the reporter line (MFI) and the number of identified interactions. The signal strength was quantified as the MFI during sorting.

The Pearson correlation coefficient is indicated in the plot. (**g**) Correlation between reporter screen hits and the TFi screen separated by reporter line (Xist^High/Xist^Neg). The Pearson correlation coefficient and the mean MFI during sorting is indicated in the panels. (**h**) Z-score transformed XX expression dynamics of activators for selected reporter lines during the RNA-seq time course (Fig. 3). TF activators are ordered via hierarchical clustering. Interaction partners with XX-biased expression are marked on the right (*ANOVA*, FDR ≤ 0.05). (**i**) Z-score transformed XX expression dynamics of TF genes activating a reporter line located in either the proximal (n = 11), distal (n = 15) or noRE-like (n = 17) PCA clusters (Fig. 4g). (**j**) As in (**i**), but comparing XX and XO expression for TF genes activating a proximal (left, n = 11) or distal (right, n = 15) reporter line (for clusters see Fig. 4g). Results of a *two-sided rank-sum Wilcoxon test* are shown. In (**i**) and (**j**) the boxes indicate the IQR, as well as the median (black line). Whiskers extend to the furthest values within 1.5 x IQR from the box.

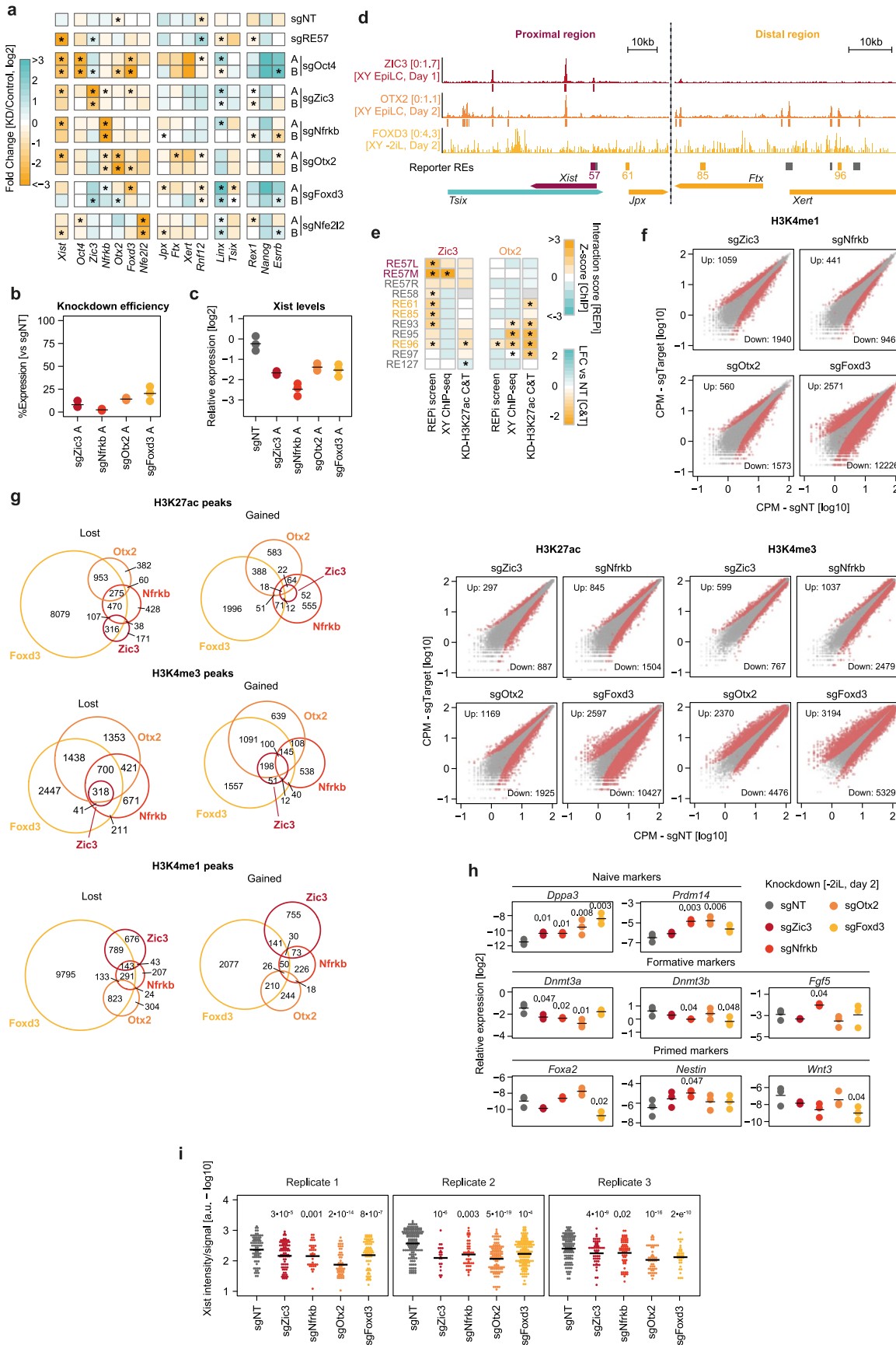

**Extended Data Fig. 9 | See next page for caption.**

**Extended Data Fig. 9 | Knock-down of selected Xist activators. (a)** Effect of knockdown of selected Xist activators via the HDAC4 CasTuner system on the expression of Xist and various Xist regulators assessed by RT-qPCR. Guides targeting RE57 were included as a positive control for Xist. The mean of 2 biological replicates for two different multiguide plasmids (A + B) per target are shown. The black asterisk denotes significance (*two-sided unpaired t-test*, p ≤ 0.05). **(b-c)** Knockdown efficiency of the indicated regulators **(b)** and the effect on Xist expression **(c)** assessed by RT-qPCR during the RNA-FISH and CUT&Tag experiment in main Fig. 5. In **(b)** the data is quantified per replicate compared to the non-targeting control. The black bar denotes the mean of three replicates. **(d)** Published ChIP-seq data, depicting binding of ZIC3, OTX2 and FOXD3 at the *Xist* locus during differentiation in XY cells. **(e)** Heatmap comparing reporter screen (REPi), ChIP-seq binding in differentiating XY cells and H3K27ac CUT&Tag upon knockdown for ZIC3 and OTX2[37,38]. The reporter screen and CUT&Tag data are quantified as interaction scores and log$_2$ fold change compared to the NTC, respectively (see Figs. 4 and 5). The ChIP-seq data is quantified as z-score transformed normalized counts across the included REs. Significant interactions from the reporter screen, REs overlapping ChIP peaks and differentially enriched peaks from the CUT&Tag analysis are marked with a black asterisk. **(f-g)** Genome-wide effect of Xist activator knockdown on selected histone marks assessed via CUT&Tag, shown as scatter plot **(f)** and Venn diagrams **(g)**. In **(f)** up- and downregulated peaks are colored in red (*Diffbind*, FDR ≤ 0.05). **(h)** Effect of Xist activator knockdown on the expression of selected pluripotency and differentiation markers assessed by RT-qPCR. Results of a *two-sided unpaired t-test* (p ≤ 0.05) are indicated. **(i)** Effect of TF gene knockdown on the RNA-FISH intensity per Xist signal, separated by biological replicate (see Fig. 5f). The black bar denotes the geometric mean per sample (21-150 Xist signals per sample). The result of a *two-sided rank-sum Wilcoxon test* (p ≤ 0.05) compared to the sgNT control is shown.

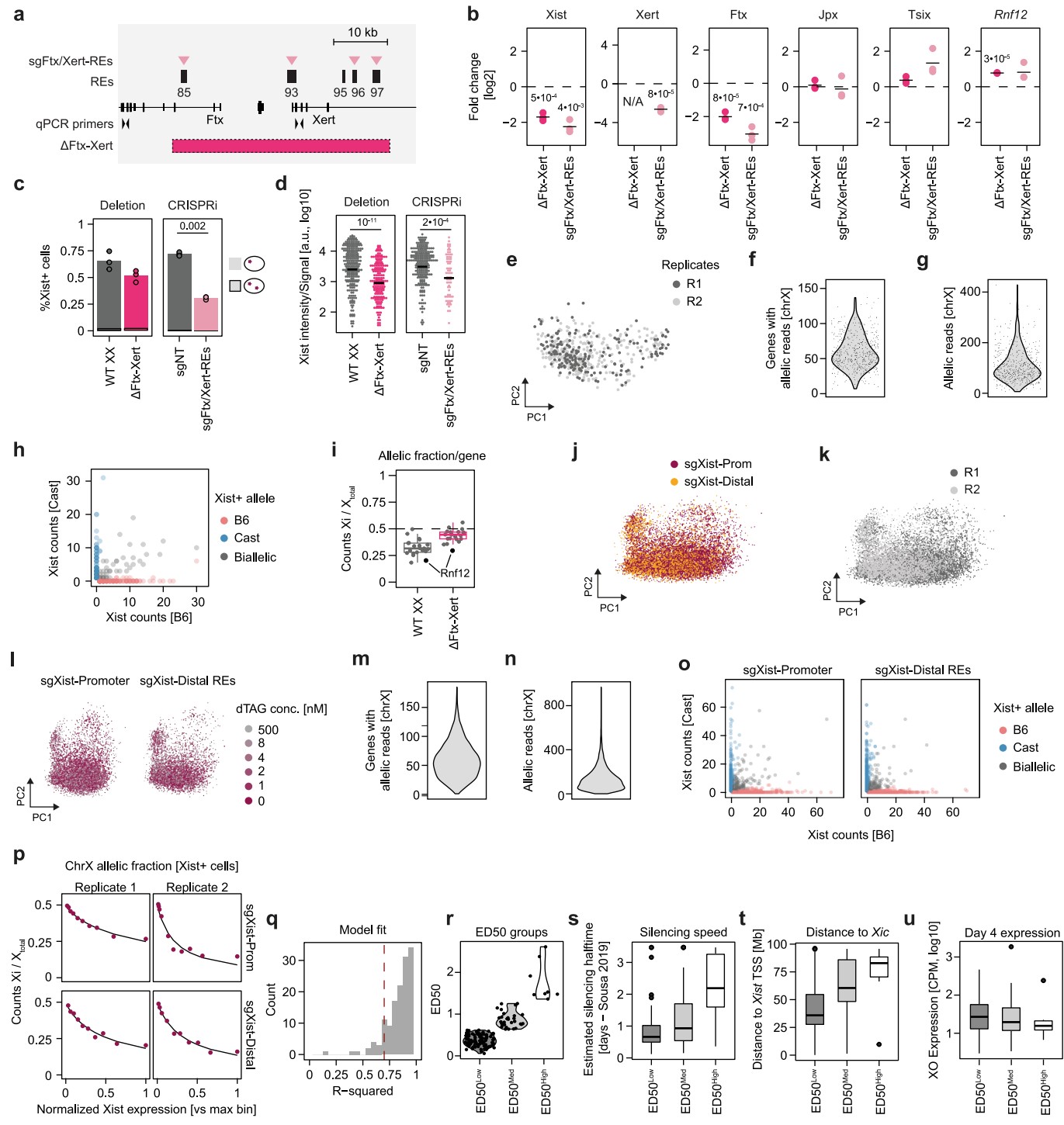

**Extended Data Fig. 10 | See next page for caption.**

**Extended Data Fig. 10 | The role of distal REs and Xist expression levels in gene silencing during XCI.** (**a-i**) Quality control measurements for the experiments in main Fig. 6a–e. (**a**) Scheme depicting the location of the ΔFtx-Xert deletion and the guide target sites used in CRISPRi. (**b**) Expression of Xist and other genes within the *Xic* at day 4 of differentiation in the ΔFtx-Xert line and following targeting of REs 85, 93, 96 and 97 by dCas9-KRAB assessed via RT-qPCR. The $\log_2$-fold change compared to the wild type line or non-targeting guides is shown. Results of a *two-sided unpaired t-test* ($p \leq 0.05$) are indicated. (**c-d**) Xist expression assessed via RNA-FISH at day 4 of differentiation in the ΔFtx-Xert line and following targeting of REs 85, 93, 96 and 97 by dCas9-KRAB (CRISPRi). Results of a *two-sided unpaired t-test* ($p \leq 0.05$) and of a *two-sided rank-sum Wilcoxon test* ($p \leq 0.05$) are indicated in (**c**) and (**d**), respectively. (**e**) PCA depicting single-cell transcriptomes, separated by replicate (502 cells). (**f**) Number of X-chromosomal genes with allelic reads per cell (502 cells). (**g**) Total number of X-chromosomal allelic reads per cell (502 cells). (**h**) Allelic Xist counts for Xist+ cells separated by allele (252 cells). Biallelic cells (Xist allelic ratio 0.2-0.8) are shown in gray and were removed from further analysis (41 cells). (**i**) Allelic ratio of X-chromosomal genes, comparing wildtype and ΔFtx-Xert cells. 25 genes with a sufficient number of allelic reads were included in the analysis. (**j-u**) Additional analyses of the experiments in Fig. 6f–o. (**j-n**) PCAs, separated by sgRNA set (**j**), replicate (**k**), and dTAG concentration (**l**), number of X-chromosomal genes with allelic reads (**m**) and total number of X-chromosomal allelic reads (**n**) per cell in the Xist titration experiment. (**o**) As in (**h**), but for the Xist titration experiment. (**p**) Dose-response analysis comparing allelic fraction of the X chromosome to normalized Xist expression, separated by cell line and replicate. The curve was fitted using a four-parameter log-logistic function. (**q**) Histogram of R-squared values describing fit of the dose-response curves to the experimental data per X-linked gene. Genes with a fit below 0.7 (red line) were excluded from further analysis. (**r**) Violin plot detailing groups of genes, separated by ED50. The fitted models were clustered into 3 groups using *kmeans*. (**s-u**) Boxplots depicting estimated silencing half-time following Xist induction (**s**), distance to *Xist* (**t**) and expression in XO cells at day 4 of differentiation (**u**). The groups contain 74 (ED50$^{Low}$), 21 (ED50$^{Mid}$) or 6 genes (ED50$^{High}$). In (**i,s-u**) the boxes indicate the IQR, as well as the median (black line). Whiskers extend to the furthest values within 1.5 x IQR from the box.

# Reporting Summary

## Statistics

For all statistical analyses, confirm that the following items are present in the figure legend, table legend, main text, or Methods section.

| n/a | Confirmed | |
|---|---|---|
| ☐ | ☒ | The exact sample size (n) for each experimental group/condition, given as a discrete number and unit of measurement |
| ☐ | ☒ | A statement on whether measurements were taken from distinct samples or whether the same sample was measured repeatedly |
| ☐ | ☒ | The statistical test(s) used AND whether they are one- or two-sided<br>*Only common tests should be described solely by name; describe more complex techniques in the Methods section.* |
| ☒ | ☐ | A description of all covariates tested |
| ☐ | ☒ | A description of any assumptions or corrections, such as tests of normality and adjustment for multiple comparisons |
| ☐ | ☒ | A full description of the statistical parameters including central tendency (e.g. means) or other basic estimates (e.g. regression coefficient) AND variation (e.g. standard deviation) or associated estimates of uncertainty (e.g. confidence intervals) |
| ☐ | ☒ | For null hypothesis testing, the test statistic (e.g. $F$, $t$, $r$) with confidence intervals, effect sizes, degrees of freedom and $P$ value noted<br>*Give P values as exact values whenever suitable.* |
| ☒ | ☐ | For Bayesian analysis, information on the choice of priors and Markov chain Monte Carlo settings |
| ☒ | ☐ | For hierarchical and complex designs, identification of the appropriate level for tests and full reporting of outcomes |
| ☐ | ☒ | Estimates of effect sizes (e.g. Cohen's $d$, Pearson's $r$), indicating how they were calculated |
| | | *Our web collection on statistics for biologists contains articles on many of the points above.* |

## Software and code

Policy information about availability of computer code

| Data collection | R (v4.2), Python (v3.10.10), MAGeCK (v0.5.9.3), opencyto (v.1.24.0), tidyverse (v1.3.2), flowCore (v1.52.1), Rsubread (v2.0.1), Samtools (v1.10), STAR (v2.7.9a), TrimGalore (v0.6.4), scikit-image (v0.22), numpy (v1.26.2), czifile (v2019.7.2), DESeq2 (v.1.42.0), fgsea (v1.24.0), FIMO (v5.1.1), bowtie2 (v2.3.5.1), bedtools (v2.29.2), Picard (v2.18.25), MACS2 (v2.2.7.1), deepTools (v3.5.5), DiffBind (v3.14.0), STARsolo (v.2.7.11a), scater (v1.26.1), samTools (v1.19.2), featureCounts (v2.0.6), Scanpy (v1.9.8), Anndata (v0.10.5), drc (v3.0.1). |
|---|---|
| Data analysis | https://github.com/EddaSchulz/TFiScreen_Paper |

For manuscripts utilizing custom algorithms or software that are central to the research but not yet described in published literature, software must be made available to editors and reviewers. We strongly encourage code deposition in a community repository (e.g. GitHub). See the Nature Portfolio guidelines for submitting code & software for further information.

## Data

Policy information about availability of data

All manuscripts must include a data availability statement. This statement should provide the following information, where applicable:
  - Accession codes, unique identifiers, or web links for publicly available datasets
  - A description of any restrictions on data availability
  - For clinical datasets or third party data, please ensure that the statement adheres to our policy

Sequencing data generated in this study is available on GEO under accession number GSE274507. Raw microscopy data is available at doi.org/10.5281/

## Research involving human participants, their data, or biological material

Policy information about studies with human participants or human data. See also policy information about sex, gender (identity/presentation), and sexual orientation and race, ethnicity and racism.

| | |
|---|---|
| Reporting on sex and gender | N/A |
| Reporting on race, ethnicity, or other socially relevant groupings | N/A |
| Population characteristics | N/A |
| Recruitment | N/A |
| Ethics oversight | N/A |

Note that full information on the approval of the study protocol must also be provided in the manuscript.

# Field-specific reporting

Please select the one below that is the best fit for your research. If you are not sure, read the appropriate sections before making your selection.

☒ Life sciences   ☐ Behavioural & social sciences   ☐ Ecological, evolutionary & environmental sciences

For a reference copy of the document with all sections, see nature.com/documents/nr-reporting-summary-flat.pdf

# Life sciences study design

All studies must disclose on these points even when the disclosure is negative.

| | |
|---|---|
| Sample size | Chosen based on experience with similar assays. Where feasible three biological replicates, which is the standard in the field, were included to confer robustness. For Xist CRISPRi screens and scRNAseq assays only two replicates were performed due to technical and financial constraints. |
| Data exclusions | For RNA-seq time course 1 repliate sample (XO_36h_R1) was excluded due to low read number. Detailed in the supplement. |
| Replication | All assays were performed with 2-3 biological replicates. Correlation between replicates is provided in supplement. All attempts at replication were successful. |
| Randomization | Not applicable, as the study only involved cell culture assays with genetically identical cells grown under uniform conditions. |
| Blinding | Not applicable, as treatments to cell culture assays were applied and analyzed by the same researcher using objective, automated measurements. |

# Reporting for specific materials, systems and methods

We require information from authors about some types of materials, experimental systems and methods used in many studies. Here, indicate whether each material, system or method listed is relevant to your study. If you are not sure if a list item applies to your research, read the appropriate section before selecting a response.

### Materials & experimental systems

| n/a | Involved in the study |
|---|---|
| ☐ | ☒ Antibodies |
| ☐ | ☒ Eukaryotic cell lines |
| ☒ | ☐ Palaeontology and archaeology |
| ☒ | ☐ Animals and other organisms |
| ☒ | ☐ Clinical data |
| ☒ | ☐ Dual use research of concern |
| ☒ | ☐ Plants |

### Methods

| n/a | Involved in the study |
|---|---|
| ☐ | ☒ ChIP-seq |
| ☐ | ☒ Flow cytometry |
| ☒ | ☐ MRI-based neuroimaging |

## Antibodies

| | |
|---|---|
| Antibodies used | H3K4me3 (Active Motif, Cat# 61379, Lot# 23107153), H3K4me1 (Active Motif, Cat# 39635, Lot# 21516012), H3K27ac (Active Motif, Cat# 39685, Lot# 23188112), Rabbit anti-mouse secondary Ab (Thermo FIsher, Cat# 31194). Antibody dilutions (1:100) are provided in Supplementary Table 6. |
| Validation | H3K4me1 and H3K27ac Abs were validated for CUT&Tag (Active Motif, https://www.activemotif.com/documents/tds/39635.pdf, https://www.activemotif.com/documents/tds/39685.pdf), H3K4me3 Ab was validated for ChIP-seq and previously published for CUT&Tag (Active Motif,https://www.activemotif.com/documents/tds/61379.pdf, https://pubmed.ncbi.nlm.nih.gov/36864748/) Secondary Ab was validated for IP applications (Thermo Fisher, https://www.thermofisher.com/order/genome-database/dataSheetPdf?producttype=antibody&productsubtype=antibody_secondary&productId=31194&version=Local) |

## Eukaryotic cell lines

Policy information about cell lines and Sex and Gender in Research

| | |
|---|---|
| Cell line source(s) | TX1072 mouse embryonic stem cells (XX and XO lines, B6/Cast, Schulz lab), Lenti-X 293T cell line (Takara, Cat# 632180) |
| Authentication | All monoclonal TX1072 lines were NGS karyotyped in-house. Lenti-X 293T line was authenticated by Takara before purchasing (https://www.takarabio.com/documents/Certificate%20of%20Analysis/632180/632180-120716.pdf). |
| Mycoplasma contamination | Confirmed negative for all TX1072 lines. Lenti-X 293T line was confirmed negative by Takara before purchasing (https://www.takarabio.com/documents/Certificate%20of%20Analysis/632180/632180-120716.pdf). |
| Commonly misidentified lines (See ICLAC register) | No commonly misidentified lines were used in this study. |

## Palaeontology and Archaeology

| | |
|---|---|
| Specimen provenance | *Provide provenance information for specimens and describe permits that were obtained for the work (including the name of the issuing authority, the date of issue, and any identifying information). Permits should encompass collection and, where applicable, export.* |
| Specimen deposition | *Indicate where the specimens have been deposited to permit free access by other researchers.* |
| Dating methods | *If new dates are provided, describe how they were obtained (e.g. collection, storage, sample pretreatment and measurement), where they were obtained (i.e. lab name), the calibration program and the protocol for quality assurance OR state that no new dates are provided.* |

☐ Tick this box to confirm that the raw and calibrated dates are available in the paper or in Supplementary Information.

| | |
|---|---|
| Ethics oversight | *Identify the organization(s) that approved or provided guidance on the study protocol, OR state that no ethical approval or guidance was required and explain why not.* |

Note that full information on the approval of the study protocol must also be provided in the manuscript.

## Animals and other research organisms

Policy information about studies involving animals; ARRIVE guidelines recommended for reporting animal research, and Sex and Gender in Research

| | |
|---|---|
| Laboratory animals | *For laboratory animals, report species, strain and age OR state that the study did not involve laboratory animals.* |
| Wild animals | *Provide details on animals observed in or captured in the field; report species and age where possible. Describe how animals were caught and transported and what happened to captive animals after the study (if killed, explain why and describe method; if released, say where and when) OR state that the study did not involve wild animals.* |
| Reporting on sex | *Indicate if findings apply to only one sex; describe whether sex was considered in study design, methods used for assigning sex. Provide data disaggregated for sex where this information has been collected in the source data as appropriate; provide overall numbers in this Reporting Summary. Please state if this information has not been collected. Report sex-based analyses where performed, justify reasons for lack of sex-based analysis.* |
| Field-collected samples | *For laboratory work with field-collected samples, describe all relevant parameters such as housing, maintenance, temperature, photoperiod and end-of-experiment protocol OR state that the study did not involve samples collected from the field.* |
| Ethics oversight | *Identify the organization(s) that approved or provided guidance on the study protocol, OR state that no ethical approval or guidance was required and explain why not.* |

Note that full information on the approval of the study protocol must also be provided in the manuscript.

# Clinical data

Policy information about clinical studies

All manuscripts should comply with the ICMJE guidelines for publication of clinical research and a completed CONSORT checklist must be included with all submissions.

Clinical trial registration | *Provide the trial registration number from ClinicalTrials.gov or an equivalent agency.*

Study protocol | *Note where the full trial protocol can be accessed OR if not available, explain why.*

Data collection | *Describe the settings and locales of data collection, noting the time periods of recruitment and data collection.*

Outcomes | *Describe how you pre-defined primary and secondary outcome measures and how you assessed these measures.*

# Dual use research of concern

Policy information about dual use research of concern

## Hazards

Could the accidental, deliberate or reckless misuse of agents or technologies generated in the work, or the application of information presented in the manuscript, pose a threat to:

| No | Yes | |
|----|-----|--|
| ☒ | ☐ | Public health |
| ☒ | ☐ | National security |
| ☒ | ☐ | Crops and/or livestock |
| ☒ | ☐ | Ecosystems |
| ☒ | ☐ | Any other significant area |

## Experiments of concern

Does the work involve any of these experiments of concern:

| No | Yes | |
|----|-----|--|
| ☒ | ☐ | Demonstrate how to render a vaccine ineffective |
| ☒ | ☐ | Confer resistance to therapeutically useful antibiotics or antiviral agents |
| ☒ | ☐ | Enhance the virulence of a pathogen or render a nonpathogen virulent |
| ☒ | ☐ | Increase transmissibility of a pathogen |
| ☒ | ☐ | Alter the host range of a pathogen |
| ☒ | ☐ | Enable evasion of diagnostic/detection modalities |
| ☒ | ☐ | Enable the weaponization of a biological agent or toxin |
| ☒ | ☐ | Any other potentially harmful combination of experiments and agents |

# Plants

Seed stocks | *Report on the source of all seed stocks or other plant material used. If applicable, state the seed stock centre and catalogue number. If plant specimens were collected from the field, describe the collection location, date and sampling procedures.*

Novel plant genotypes | *Describe the methods by which all novel plant genotypes were produced. This includes those generated by transgenic approaches, gene editing, chemical/radiation-based mutagenesis and hybridization. For transgenic lines, describe the transformation method, the number of independent lines analyzed and the generation upon which experiments were performed. For gene-edited lines, describe the editor used, the endogenous sequence targeted for editing, the targeting guide RNA sequence (if applicable) and how the editor was applied.*

Authentication | *Describe any authentication procedures for each seed stock used or novel genotype generated. Describe any experiments used to assess the effect of a mutation and, where applicable, how potential secondary effects (e.g. second site T-DNA insertions, mosiacism, off-target gene editing) were examined.*

# ChIP-seq

## Data deposition

☒ Confirm that both raw and final processed data have been deposited in a public database such as GEO.

☐ Confirm that you have deposited or provided access to graph files (e.g. BED files) for the called peaks.

| | |
|---|---|
| Data access links<br>*May remain private before publication.* | GSE273068 |
| Files in database submission | Bigwig, fastq |
| Genome browser session<br>(e.g. UCSC) | https://genome.ucsc.edu/s/tillsc23/CnT_xist_act |

## Methodology

| | |
|---|---|
| Replicates | 3 |
| Sequencing depth | paired-end, 100 bp, 3.5-9.3*10^6 fragments (CUT&Tag) |
| Antibodies | H3K4me3 (Active Motif, Cat# 61379, Lot# 23107153), H3K4me1 (Active Motif, Cat# 39635, Lot# 21516012), H3K27ac (Active Motif, Cat# 39685, Lot# 23188112), Rabbit anti-mouse secondary Ab (Thermo FIsher, Cat# 31194) - Used for CUT&Tag |
| Peak calling parameters | MACS2 standard options |
| Data quality | Nr. of peaks, FrIP and duplicate reads available in supplement |
| Software | https://github.com/EddaSchulz/TFiScreen_Paper |

# Flow Cytometry

## Plots

Confirm that:

☒ The axis labels state the marker and fluorochrome used (e.g. CD4-FITC).

☒ The axis scales are clearly visible. Include numbers along axes only for bottom left plot of group (a 'group' is an analysis of identical markers).

☒ All plots are contour plots with outliers or pseudocolor plots.

☒ A numerical value for number of cells or percentage (with statistics) is provided.

## Methodology

| | |
|---|---|
| Sample preparation | Mouse embryonic stem cells, unfixed with GFP reporter (Reporter screens) or fixed with PrimeFlow kit (Xist screens) |
| Instrument | FACSAria Fusion flow cytometer (BD) |
| Software | https://github.com/EddaSchulz/TFiScreen_Paper |
| Cell population abundance | Not applicable (sorted cells were harvested for CRISPR screens) |
| Gating strategy | Gating live cells + singlets with forward and sideward scatters, then sorting GFP/Alexa 647 high and low populations |

☒ Tick this box to confirm that a figure exemplifying the gating strategy is provided in the Supplementary Information.

# Magnetic resonance imaging

## Experimental design

| | |
|---|---|
| Design type | *Indicate task or resting state; event-related or block design.* |
| Design specifications | *Specify the number of blocks, trials or experimental units per session and/or subject, and specify the length of each trial or block (if trials are blocked) and interval between trials.* |
| Behavioral performance measures | *State number and/or type of variables recorded (e.g. correct button press, response time) and what statistics were used to establish that the subjects were performing the task as expected (e.g. mean, range, and/or standard deviation across* |

*(subjects).*

## Acquisition

Imaging type(s)
*Specify: functional, structural, diffusion, perfusion.*

Field strength
*Specify in Tesla*

Sequence & imaging parameters
*Specify the pulse sequence type (gradient echo, spin echo, etc.), imaging type (EPI, spiral, etc.), field of view, matrix size, slice thickness, orientation and TE/TR/flip angle.*

Area of acquisition
*State whether a whole brain scan was used OR define the area of acquisition, describing how the region was determined.*

Diffusion MRI  ☐ Used  ☐ Not used

## Preprocessing

Preprocessing software
*Provide detail on software version and revision number and on specific parameters (model/functions, brain extraction, segmentation, smoothing kernel size, etc.).*

Normalization
*If data were normalized/standardized, describe the approach(es): specify linear or non-linear and define image types used for transformation OR indicate that data were not normalized and explain rationale for lack of normalization.*

Normalization template
*Describe the template used for normalization/transformation, specifying subject space or group standardized space (e.g. original Talairach, MNI305, ICBM152) OR indicate that the data were not normalized.*

Noise and artifact removal
*Describe your procedure(s) for artifact and structured noise removal, specifying motion parameters, tissue signals and physiological signals (heart rate, respiration).*

Volume censoring
*Define your software and/or method and criteria for volume censoring, and state the extent of such censoring.*

## Statistical modeling & inference

Model type and settings
*Specify type (mass univariate, multivariate, RSA, predictive, etc.) and describe essential details of the model at the first and second levels (e.g. fixed, random or mixed effects; drift or auto-correlation).*

Effect(s) tested
*Define precise effect in terms of the task or stimulus conditions instead of psychological concepts and indicate whether ANOVA or factorial designs were used.*

Specify type of analysis:  ☐ Whole brain  ☐ ROI-based  ☐ Both

Statistic type for inference
*Specify voxel-wise or cluster-wise and report all relevant parameters for cluster-wise methods.*

(See Eklund et al. 2016)

Correction
*Describe the type of correction and how it is obtained for multiple comparisons (e.g. FWE, FDR, permutation or Monte Carlo).*

## Models & analysis

| n/a | Involved in the study |
|-----|-----------------------|
| ☐ | ☐ Functional and/or effective connectivity |
| ☐ | ☐ Graph analysis |
| ☐ | ☐ Multivariate modeling or predictive analysis |

Functional and/or effective connectivity
*Report the measures of dependence used and the model details (e.g. Pearson correlation, partial correlation, mutual information).*

Graph analysis
*Report the dependent variable and connectivity measure, specifying weighted graph or binarized graph, subject- or group-level, and the global and/or node summaries used (e.g. clustering coefficient, efficiency, etc.).*

Multivariate modeling and predictive analysis
*Specify independent variables, features extraction and dimension reduction, model, training and evaluation metrics.*

