## [Peer Review File · Nature Structural & Molecular Biology]

Reporter CRISPR screens decipher cis- and trans-regulatory principles at the Xist locus

Corresponding Author: Dr Edda Schulz

Version 0:

Decision Letter:

23rd Oct 2024

Dear Dr. Schulz,

Thank you again for submitting your manuscript "Reporter CRISPR screens decipher cis- and trans-regulatory principles at the Xist locus". I apologise for the delay in responding, which resulted from the difficulty in timely obtaining suitable referee reports. Nevertheless, we now have comments (below) from the 3 reviewers who evaluated your paper. In light of these reports, we remain interested in your study and would like to see your response to the comments of the referees, in the form of a revised manuscript.

You will see that though the experts are appreciative of the functional insight into Xist regulation, they raise concerns on the clarity of the message and the need for further analyses/experiments to boost the value of the manuscript. More specifically, both Reviewer #1 and #2 note that the first part of the manuscript, while comprehensive, is in need of streamlining and improved clarity to help the reader. Furthermore, Reviewers #2 and #3 request new ChIP-Seq data or analyses of existing ones to understand where the newly identified TFs bind. We editorially agree that such experiments/analyses, combined with all the requested controls and addressing the questions raised about the potential generalisability of the findings to loci other than Xist by the Reviewers, will significantly improve the manuscript and therefore kindly ask you to follow the guidelines of the referees towards that end. Please be sure to address/respond to all concerns of the referees in full in a point-by-point response and highlight all changes in the revised manuscript text file. If you have comments that are intended for editors only, please include those in a separate cover letter.

We expect to see your revised manuscript within 3-4 months. If you cannot send it within this time, please contact us to discuss an extension; we would still consider your revision, provided that no similar work has been accepted for publication at NSMB or published elsewhere.

Reporting Summary:

Data availability: this journal strongly supports public availability of data. All data used in accepted papers should be available via a public data repository, or alternatively, as Supplementary Information. If data can only be shared on request, please explain why in your Data Availability Statement, and also in the correspondence with your editor. Please note that for some data types, deposition in a public repository is mandatory - more information on our data deposition policies and available repositories can be found below:
<https://www.nature.com/nature-research/editorial-policies/reporting-standards#availability-of-data>

Nature Structural & Molecular Biology is committed to improving transparency in authorship. As part of our efforts in this direction, we are now requesting that all authors identified as 'corresponding author' on published papers create and link their Open Researcher and Contributor Identifier (ORCID) with their account on the Manuscript Tracking System (MTS), prior to acceptance. This applies to primary research papers only. ORCID helps the scientific community achieve unambiguous attribution of all scholarly contributions. You can create and link your ORCID from the home page of the MTS by clicking on 'Modify my Springer Nature account'. For more information please visit please visit www.springernature.com/orcid.

Link Redacted

Sincerely,

Dimitris Typas
Senior Editor
Nature Structural & Molecular Biology
ORCID: 0000-0002-8737-1319

Reviewers' Comments:

Reviewer #1 (Remarks to the Author):

The manuscript by Schwaemmle and colleagues sets the stage with a number of CRISPR screens identifying TFs that regulate Xist along an cellular model of mouse differentiation. Complimentary to earlier work by the Schulz group, the screens unravel several key TFs regulating distinct aspects of Xist regulation along differentiation (e.g. Oct4). The manuscript then continues with a combined RE-TF screen (Fig. 4) that reveals early and late regulators of Xist acting with distinct modes. Zic3, Nrkb, Otx2 and Foxd3 are then dissected further using chromatin-state analysis along differentiation as well as Xist regulation (by RNA-FISH). Lastly, the authors use a Ftx deletion cell line to dissect the impact of distal REs on Xist levels and X inactivation. The overall conclusion of the paper is that basal Xist activation and high expression levels can be controlled independently from each other via distal and proximal REs.

I congratulate the authors on the impressive work done here. The conclusions are well supported using orthogonal methodology and basically all controls are presented. I see no major experimental shortcomings, statistics are always

provided, the data on GEO is complete and all bioinformatic codes present on github.

Comments:

The manuscript contains a lot of high quality data already and I acknowledge the complexity (and beauty!) of the experimental designs. However, it took me quite a lot of time (more than usual) to understand what has actually been done. For example, the introduction paragraph nicely delineates that the paper aims to understand connections between TFs and REs. But then this part is only really presented in Fig. 4. I understand that the Figs. 1-3 and the corresponding screen have been a lot of work and are an important foundation for what is presented in the second half of the paper. However, the results (although comprehensively done) are not very groundbreaking. In the interest of a “general” reader, I would recommend the authors to shorten this part, perhaps even condensing Fig.1-3 into a single Figure focusing only on the very essentials (which TFs and then a bit of dissection on Oct4). Furthermore, the schemes such as the ones in Fig. 1a could be simplified (I got confused by “911 promoters” and initially thought that the screen is targeting 911 regions nearby Xist and 602 TF genes; what do the symbols above “selected” and “unsorted” mean? Alexa-647 is another irrelevant detail). For Figs 1-3 I don't think the details matter so much – basically you are performing genome-wide screens to identify regulators of Xist along differentiation.

Besides that, in the discussion paragraph the authors should expand a bit on the fact that what is studied here is a non-coding RNA present in the nucleus (i.e. doesn't encode a protein). What could the findings mean if the target molecule was a protein-coding gene (that could in turn be regulated by protein turnover and/or signalling)? And what if the target molecule was a transcription factor itself?

Extended Data:

I understand the effort for brevity, but from the main text it is not very easy to grasp the data in most Extended Data Figs., but especially the EDs 1-4. The legends do not help either (for example, what does it mean “Significance was assessed using MAGeCK mle” or “The score is given by the fold change between the 90th and the 10th percentile.”). I understand these are mostly QC metrics and additional analyses for the screen. Nonetheless they should be understandable. Here, I recommend preparing a Supplementary Note with more careful explanation of what those EDs mean and be available for the interested reader.

Reviewer #2 (Remarks to the Author):

This study takes a systematic approach to identify TFs and their cognate REs for the control of Xist gene expression in mouse XX embryonic stem cells undergoing differentiation. CRISPRi screens identified several TFs that function either as Xist activators or repressors. A high-throughput assay for Xist expression levels after 2 days of differentiation identified hits resulting in high (no effect), low (inferred role in boosting Xist levels) or zero (inferred effect on initiating basal Xist expression). Hits included known factors (notably known Xist repressors in pluripotent cells) and also several novel TFs. Oct4, previously characterised as an Xist repressor in undifferentiated cells was found to have an activating role during differentiation. Profiling mRNAs levels of genes encoding identified hits over the differentiation timecourse identified subsets of TFs expressed transiently at higher levels either during early differentiation stages or somewhat later over the differentiation timecourse and these were inferred to be involved respectively in basal Xist upregulation or maintaining/boosting levels over time. Several of the TF hits were expressed at a higher level in XX cells (determined from comparison with X0 cell line), a feature more predominant for the TFs implicated in basal Xist upregulation. Selected factors were analysed further, notably using a novel reporter screen approach to define whether they acted through previously defined RE elements lying proximal or distal to Xist. Intriguingly, novel TFs implicated in basal Xist expression tended to associate with proximal REs, whereas TFs implicated in boosting Xist expression associated with more with distal REs. This observation leads the authors to conclude that proximal and distal REs have distinct roles, contributing to a two-step process to initiate and then sustain physiological Xist expression levels. To investigate the importance of boosting transcription via distal elements the authors analyse X-linked gene silencing in an XX ES cell model with a large deletion encompassing several distal REs and in which Xist is expressed in a high proportion of cells but at reduced levels. This analysis revealed that gene silencing efficiency is indeed reduced linked, suggesting that achieving high levels of Xist expression is important for proper X chromosome silencing.

Overall this is a comprehensive and high-quality study that develops and applies novel and ambitious systematic approaches to dissect the regulatory circuitry of a biologically important process, developmental regulation of Xist RNA expression. The study provides new advances towards understanding Xist gene regulation, notably in defining novel activating TFs with and inferring the two-step mechanism to achieve full levels of Xist expression. I do have some comments/questions as follows:

1. Depleting transcription factors could impact Xist expression directly or in more indirect ways, for example affecting the differentiation trajectory of the cells. Direct effects were inferred (Fig E9e) but not directly tested. I was wondering if further evidence for direct roles could be provided using ChIP-seq analysis for one or more of the novel TFs? Mutating binding sites within an RE in the native locus would provide the ultimate proof for direct function, although I recognise that this is a major undertaking.
2. Prior work has identified YY1 as a key TF that functions through the exon 1 enhancer to promote Xist upregulation yet it hasn't come through in CRISPR screens herein despite being included as an added control for the CasTuner screen. Additionally, depletion of novel basal TFs like Zic1 suggested to bind to the exon 1 enhancer, largely abrogates Xist upregulation. Do the authors envisage cooperative effects of YY1 and the novel TFs or something else? Including discussion around this point may help readers to relate this work with prior studies on mouse Xist gene regulation.
3. Along the same lines, a recent paper from the authors defined a key role for specific GATA family TFs in sustaining Xist expression in early mouse embryos, but these TFs did not come through as hits in the CRISPR screen. It would be good to include discussion around this point to help readers to integrate findings from the two papers?
4. The three transient expression sub-groupings in Figure 3 look very similar to my eye and I wasn't clear how transient group 2 and 3 patterns are consistent with booster TFs being needed for ongoing Xist expression, beyond 2-3 days of

differentiation. Some clarification on this point would be helpful.

5. The large deletion used for the experiment in Fig 6 results in removal of distal REs but also rearranges the locus, possibly impacting Xist expression by bringing together sequences that are normally far apart or changing expression of upstream ncRNAs that regulate Xist in cis. Can the authors comment on the degree to which effects from this deletion can be attributed solely to loss of specific REs vs other factors?

6. As a non-systems biology reader I found it challenging to follow and digest the large amount of data and analysis described in this paper. It may be useful to include some more simplified summary points after specific experiments/analyses to improve readability.

Reviewer #3 (Remarks to the Author):

In this manuscript, the authors develop a combined CRISPR-based screening approach to identify transcription factors (TFs) that regulate the long non-coding RNA Xist and map the interactions between TFs and regulatory elements. Their goal is to clarify the relationship between TFs and their regulatory elements and to better understand the complex interdependencies involved. They reveal a group of transiently expressed TFs that regulate nearby regulatory elements (REs), driving the step-wise activation of Xist expression.

This work addresses a significant and still poorly understood question in the field of X-chromosome inactivation (XCI)—how cells determine the levels of Xist activation. It describes novel regulators of that process and links them with their regulatory sequence near Xist. Moreover, the manuscript is clearly written with well-structured and easy-to-follow figures.

Major comments:

1) Oct4 function

- The authors propose a dual role for Oct4 in both activating and repressing Xist, which is an interesting observation but requires additional supporting data and clarification. For instance, where does Oct4 bind on the X chromosome in differentiated versus undifferentiated states?

- It seems that in the RA conditions, Oct4 knockdown was not very effective (Figure 2). Could this be why Xist levels showed little change? Repeating the RT-qPCR experiment might help confirm the reliability of these results.

- Furthermore, the statement "rapid OCT4 downregulation during RA differentiation is associated with reduced Xist levels compared to 2iL withdrawal" is unclear in the context of Figure 2 and needs further explanation.

2) Systems used in the manuscript:

- Is Xist expressed in the XO system? Based on Figure 3c, it seems that Xist is not expressed, so how do the authors determine that the observed response is due to chromosome number recognition rather than the presence of Xist? Additionally, what does the expression profile of activators and repressors look like in XY cells? At this point, I disagree with the manuscript's repeated claim that the number of X chromosomes is the primary cue.

- It is also unclear how the authors control for X chromosome loss, which frequently occurs in XX cell cultures and could potentially lead to false scoring as Xist-negative. How is this accounted for in their analysis?

3) Further characterization of identified transcription factors (TFs):

- While I appreciate the screening approach and the use of previously identified regulatory elements (REs), I wonder why the authors did not generate or analyze previously available ChIP-seq (or similar data) for the binding sites of the identified factors. Providing at least an in-silico analysis of the binding sites for these TFs would strengthen the conclusions.

- What happens in vivo when one of the identified factors (Zic3, Nfrkb, Otx2, Foxd3) is malfunctioning? Does X-chromosome inactivation (XCI) still progress normally, and are the animals viable?

4) Is there a minimal threshold of Xist activity that allows proper XCI? It seems to me that authors are able to address this question using their RE-TF interplay. It will significantly add to the mechanistical understanding of the dual Xist activation (why and when).

5) As authors commence their intro focusing on a general gene regulatory question, I wonder if distal/proximal function of Zic3, Nfrkb, Otx2, Foxd3 is something that these factors exhibit in general (i.e. their other targets) or happens only during XCI and for Xist. Their non-X targets would be interesting to explore/provide in the context of these observations. Similarly, how scalable are the observations to other ncRNAs or coding genes? Xist is a very specific locus like no other in a mammalian genome, thus, its regulation might not be easily translatable to other loci.

Minor

- In the undifferentiated state, where Oct4 acts as a repressor, its effects seem much stronger than in its later role as an activator. The authors should address this difference, considering both the direct and indirect effects of Oct4 are proposed.
- It would be helpful to explain how the FIREWCh system works, unfortunately Fig 3a is not helpful in that sense as it is unclear what binds to what and upon which signal.
- How do authors decide to use 2iL withdrawal as their experimental model for differentiation and not RA or Fgf2/AcA protocols? It should be explained.
- Was the Xist detection range (by FISH or other means) comparable in the two screens?
- If the second screen is performed after longer time, wouldn't it deliver different group of regulators (mainly those acting in later stages of differentiation)? I might have missed it but it would be good to explain in case that's not the expectation.
- In Figure 2 it seems like there's a panel missing with the actual 2nd screen results. I see it in the supplement, but it would be great to have it in the main. Can authors provide the tracks from sorting?
- Out of curiosity, wouldn't RNA decay mechanisms that regulate stability of Xist affect results/conclusions of the screens? It seems like an important mechanism that is linked with the need to boost Xist expression at certain differentiation time-point.

Version 1:

Decision Letter:

Our ref: NSMB-A49764A

9th May 2025

Dear Dr. Schulz,

Thank you for submitting your revised manuscript "Reporter CRISPR screens decipher cis- and trans-regulatory principles at the Xist locus" (NSMB-A49764A). It has now been seen by the original referees and their comments are below. The reviewers find that the paper has improved in revision and retain no concerns, therefore we can accept the manuscript in principle in Nature Structural & Molecular Biology, pending minor revisions to comply with our editorial and formatting guidelines.

We are now performing detailed checks on your paper and will send you a checklist detailing our editorial and formatting requirements in about two weeks. Please do not upload the final materials and make any revisions until you receive this additional information from us.

To facilitate our work at this stage, it is important that we have a copy of the main text as a word file. If you could please send along a word version of this file as soon as possible, we would greatly appreciate it; please make sure to copy the NSMB account (cc'ed above).

Sincerely,

Dimitris Typas
Senior Editor
Nature Structural & Molecular Biology
ORCID: 0000-0002-8737-1319

Reviewer #1 (Remarks to the Author):

I have read the replies to the reviewers. I think the concerns have been sufficiently addressed and the paper can be published.

Reviewer #2 (Remarks to the Author):

The authors have provided a thorough response to my comments and have added new experiments/analyses that address the key points that I raised.

Reviewer #3 (Remarks to the Author):

The revised manuscript is much clearer and easier to follow. Authors also addressed my comments and added necessary clarifications, analysis, or removed questionable parts. I do not have any more comments that require follow up.

Version 2:

Decision Letter:

1st Aug 2025

Dear Dr. Schulz,

We are now happy to accept your revised paper "Reporter CRISPR screens decipher cis- and trans-regulatory principles at the Xist locus" for publication as an Article in Nature Structural & Molecular Biology.

Your paper will be published online soon after we receive proof corrections and will appear in print in the next available issue. You can find out your date of online publication by contacting the production team shortly after sending your proof corrections.

Authors may need to take specific actions to achieve compliance with funder and institutional open access mandates. If your research is supported by a funder that requires immediate open access (e.g. according to [Plan S principles](https://www.springernature.com/gp/open-science/plan-s-compliance) or the [NIH public access policy](https://www.springernature.com/gp/open-science/us-federal-agency-compliance)) then you should select the gold OA route, and we will direct you to the compliant route where possible. Because authors warrant under our subscription licensing terms that they haven't committed to licensing any version of their article under a licence inconsistent with the terms of our agreement – including the applicable embargo period – publication under the subscription model isn't suitable for authors whose funders require no embargo.

Sincerely,

Dimitris Typas
Senior Editor
Nature Structural & Molecular Biology
ORCID: 0000-0002-8737-1319

Reviewer 1

1. The manuscript contains a lot of high quality data already and I acknowledge the complexity (and beauty!) of the experimental designs. However, it took me quite a lot of time (more than usual) to understand what has actually been done. For example, the introduction paragraph nicely delineates that the paper aims to understand connections between TFs and REs. But then this part is only really presented in Fig. 4. I understand that the Figs. 1-3 and the corresponding screen have been a lot of work and are an important foundation for what is presented in the second half of the paper. However, the results (although comprehensively done) are not very groundbreaking. In the interest of a “general” reader, I would recommend the authors to shorten this part, perhaps even condensing Fig. 1-3 into a single Figure focusing only on the very essentials (which TFs and then a bit of dissection on Oct4). Furthermore, the schemes such as the ones in Fig. 1a could be simplified (I got confused by “911 promoters” and initially thought that the screen is targeting 911 regions nearby Xist and 602 TF genes; what do the symbols above “selected” and “unsorted” mean? Alexa-647 is another irrelevant detail). For Figs 1-3 I don’t think the details matter so much – basically you are performing genome-wide screens to identify regulators of Xist along differentiation.

We thank the reviewer for their enthusiasm and the constructive feedback on accessibility of the manuscript. We have significantly streamlined the text associated with Fig. 1-3. Some of the technical details are now discussed in Supplementary Note 1 (Screen quality controls) and in Supplementary Note 2 (Screen results). Although we decided to keep three separate figures, we tried to simplify the schematic figure 1a, describing the screen setup.

2. Besides that, in the discussion paragraph the authors should expand a bit on the fact that what is studied here is a non-coding RNA present in the nucleus (i.e. doesn’t encode a protein). What could the findings mean if the target molecule was a protein-coding gene (that could in turn be regulated by protein turnover and/or signalling)? And what if the target molecule was a transcription factor itself?

Since Xist is a Pol II transcribed spliced transcript, similar to protein-coding genes, transcriptional control mechanisms likely follow similar principles. Nevertheless, additional layers of control Xist for both coding and non-coding transcripts. We have added a small section about this point to the discussion. Moreover, we discuss in Supplementary Note 2 the fact that some screen hits might modulate post-transcriptional control of Xist.

3. Extended Data: I understand the effort for brevity, but from the main text it is not very easy to grasp the data in most Extended Data Figs., but especially the EDs 1-4. The legends do not help either (for example, what does it mean “Significance was assessed using MAGeCK mle” or “The score is given by the fold change between the 90th and the 10th percentile.”). I understand these are mostly QC metrics and additional analyses for the screen. Nonetheless they should be understandable. Here, I recommend preparing a Supplementary Note with more careful explanation of what those EDs mean and be available for the interested reader.

We are very happy to follow the reviewer’s suggestion to prepare a supplementary note describing the QC metrics in more detail (Supplementary Note 1). In addition, we re-organized Extended Data Figs. 1-4 into three figures (Extended Data Fig. 1-3). Lastly, we have added descriptive headlines to many figure panels in Extended Data Figs. 1-3 and 7 to improve the clarity of the presented analyses.

Reviewer 2

1. Depleting transcription factors could impact Xist expression directly or in more indirect ways, for example affecting the differentiation trajectory of the cells. Direct effects were inferred (Fig E9e) but not directly tested. I was wondering if further evidence for direct roles could be provided using ChIP-seq analysis for one or more of the novel TFs? Mutating binding sites within an RE in the native locus would provide the ultimate proof for direct function, although I recognise that this is a major undertaking.

The question of whether the detected interactions are direct or not is indeed highly relevant. We have performed additional analyses to (1) assess whether TF knock-down affects differentiation (Extended Data Fig. 9h) and to test direct TF binding (Extended Data Fig. 9d-e). These analyses revealed that in particular FOXD3 seems to control Xist in an indirect manner, since knock-down had pronounced genome-wide effects by interference with differentiation, and we could not detect binding to any of the tested REs in a published ChIP-seq data set. For ZIC3, OTX2 and OCT4 by contrast we could detect binding to REs in accordance with the screen results.

2. Prior work has identified YY1 as a key TF that functions through the exon 1 enhancer to promote Xist upregulation yet it hasn't come through in CRISPR screens herein despite being included as an added control for the CasTuner screen. Additionally, depletion of novel basal TFs like Zic3 suggested to bind to the exon 1 enhancer, largely abrogates Xist upregulation. Do the authors envisage cooperative effects of YY1 and the novel TFs or something else? Including discussion around this point may help readers to relate this work with prior studies on mouse Xist gene regulation.

We agree with the reviewer that it is surprising that we do not identify YY1 in our screens. The reason might be inefficiency of the Yy1-targeting guides in our sgRNA libraries. While Yy1 was not detected, our screens revealed NFRKB as a strong Xist activator, which interacts with YY1 in the context of the INO80 complex (PMID: 19062292). As pointed out by the reviewer, both ZIC3 and NFRKB/YY1 activate basal Xist expression via RE57 (exon1 enhancer). The 25% of cells that still upregulate Xist upon knock-down of Zic3, might be attributable to the YY1/NFRKB axis. The relative roles of these two pathways remains an exciting question for future studies, which we now point out in the discussion. We also now cover the limitations of the screen and interpretation of the results in more detail in Supplementary note 2.

3. Along the same lines, a recent paper from the authors defined a key role for specific GATA family TFs in sustaining Xist expression in early mouse embryos, but these TFs did not come through as hits in the CRISPR screen. It would be good to include discussion around this point to help readers to integrate findings from the two papers?

We recently described GATA family TFs as important activators of Xist transcription during imprinted XCI (PMID: 37932452). However, GATA TFs are not expressed at the onset of random XCI *in vivo* or *in vitro* and thus do not regulate Xist at this stage. Consequently, they are not detected in our loss-of-function screens. We now discuss this point in Supplementary Note 2.

4. The three transient expression sub-groupings in Figure 3 look very similar to my eye and I wasn't clear how transient group 2 and 3 patterns are consistent with booster TFs being needed for ongoing Xist expression, beyond 2-3 days of differentiation. Some clarification on this point would be helpful.

As pointed out by the reviewer the dynamics of the Transient Clusters are similar. To clarify the differences we have now indicated the time point of peak expression in Fig. 3d. It is indeed an intriguing question, why so many activators are only transiently upregulated, but expression of the target (Xist) is sustained. Ultimately, our data cannot answer this question, because the screens are all performed at an early time point of Xist upregulation (day 2). It is therefore possible that we miss factors involved in maintenance of Xist expression. However, many of the identified activators remain expressed at later time points albeit at reduced levels (Extended Data Fig. 5d), and might also contribute to maintenance of Xist expression. We now discuss this question in the discussion section.

5. The large deletion used for the experiment in Fig 6 results in removal of distal REs but also rearranges the locus, possibly impacting Xist expression by bringing together sequences that are normally far apart or changing expression of upstream ncRNAs that regulate Xist in cis. Can the authors comment on the degree to which effects from this deletion can be attributed solely to loss of specific REs vs other factors?

To verify that the observed reduction of Xist expression is can indeed be attributed to the loss of the REs within Xert and Ftx, we have performed a dedicated experiment to directly compare the dFtx-Xert mutant to a CRISPRi line targeting REs 85, 93, 96 and 97, using our dCas9-KRAB system. Although the CRISPRi perturbation exhibits a stronger effect on the number Xist-positive cells, both perturbations result in similarly reduced Xist RNA levels within Xist+ cells. This result demonstrates that the REs within the deleted region indeed function to boost Xist levels. These new data are shown in Extended Data Fig. 10.

6. As a non-systems biology reader I found it challenging to follow and digest the large amount of data and analysis described in this paper. It may be useful to include some more simplified summary points after specific experiments/analyses to improve readability.

We appreciate the reviewer's feedback on readability. We have expanded some of the summary paragraphs and reduced technical details to improve readability of the manuscript. We have further included a summary figure with the discussion (Figure 7).

Reviewer #3

Major comments:

1a. The authors propose a dual role for Oct4 in both activating and repressing Xist, which is an interesting observation but requires additional supporting data and clarification. For instance, where does Oct4 bind on the X chromosome in differentiated versus undifferentiated states?

We agree that the observed dual role of Oct4 in Xist regulation is intriguing, but somewhat puzzling from a mechanistic point of view. Given Oct4's central function in controlling cell fate transitions during early development, it is particularly challenging to disentangle direct and indirect effects. To assess OCT4 binding to the Xist locus, we have re-analyzed a published data set (now shown in Fig. 2e). In agreement with our results, Oct4 binds several Xist enhancers, specifically upon differentiation, suggesting that the observed activating function is at least in part direct. Although we cannot exclude a direct repressive function mediated by other REs in undifferentiated cells, an alternative explanation would be that Xist upregulation upon Oct4 knock-down is triggered by dedifferentiation to trophectoderm, as previously reported. Dedifferentiation is accompanied by upregulation of several GATA TF, which are potent Xist activators. We discuss this question now in more detail in the results section.

1b. It seems that in the RA conditions, Oct4 knockdown was not very effective (Figure 2). Could this be why Xist levels showed little change? Repeating the RT-qPCR experiment might help confirm the reliability of these results. Furthermore, the statement "rapid OCT4 downregulation during RA differentiation is associated with reduced Xist levels compared to 2iL withdrawal" is unclear in the context of Figure 2 and needs further explanation.

As pointed out by the reviewer Oct4 knockdown was inefficient during RA differentiation. Also repeating the qPCR did not change this observation. We have therefore decided to remove the RA differentiation from the manuscript.

2a) Is Xist expressed in the XO system? Based on Figure 3c, it seems that Xist is not expressed, so how do the authors determine that the observed response is due to chromosome number recognition rather than the presence of Xist? Additionally, what does the expression profile of activators and repressors look like in XY cells? At this point, I disagree with the manuscript's repeated claim that the number of X chromosomes is the primary cue.

We agree with the reviewer that effects of X-chromosome number could be fairly indirect. We treat it as the primary determinant because it is the only genetic difference between our XX and XO cell lines. However, the downstream effects might be mediated by genes whose expression responds to X-chromosome number. One of these could be Xist as pointed out by the reviewer. However, given that differences in expression of XX-biased Xist activators (Fig. 3) are observed prior to Xist-mediated genes silencing (day 1 of differentiation), these differences are likely not mediated by Xist. To assess whether XX-biased expression is also observed in a male-female comparison, we reanalyzed published scRNA-seq data set of peri-implantation development in vivo. Similar to our in vitro XX/XO system we observe XX-biased expression of Xist activators in cluster "Transient 1" at E4.5, a time point shortly before Xist upregulation, which occurs at E5.0. This new analysis is shown in Extended Data Fig. 5f. In addition, we now point out in the discussion that X-chromosome number effects could be mediated by various indirect mechanisms.

2b) It is also unclear how the authors control for X chromosome loss, which frequently occurs in XX cell cultures and could potentially lead to false scoring as Xist-negative. How is this accounted for in their analysis?

We regularly confirm XX status in all cell lines generated and prior to each experiment. For all reported experiments the fraction of XX cells surpassed 90%. The results are now listed in Supplementary Table 6. In the sc-RNAseq experiments, XX status was assessed via allelic reads originating from the X chromosomes. In all replicates the percentage of XX cells surpassed 97%. The results are now listed in Supplementary Table 5.

3a) Further characterization of identified transcription factors (TFs): While I appreciate the screening approach and the use of previously identified regulatory elements (REs), I wonder why the authors did not generate or analyze previously available ChIP-seq (or similar data) for the binding sites of the identified factors. Providing at least an in-silico analysis of the binding sites for these TFs would strengthen the conclusions.

We are happy to follow the reviewer's suggestion and have included a reanalysis of published ChIP-seq data sets in the revised version of the manuscript (Extended Data Fig. 9d-e). This analysis shows that, while effects of FOXD3 are likely indirect, the binding pattern of ZIC3 and OTX2 largely agrees with our screening results.

3b) What happens in vivo when one of the identified factors (Zic3, Nfrkb, Otx2, Foxd3) is malfunctioning? Does X-chromosome inactivation (XCI) still progress normally, and are the animals viable?

While, to our knowledge, Nfrkb has not been studied *in vivo*, knockout mouse models exist for the other three genes. While their loss is associated with strong developmental phenotypes, the progression of XCI has not been studied in either of them. Homozygous *Otx2* knockout is embryonic lethal around E10 (PMID: 7590242), while *Foxd3*^{-/-} mice die at gastrulation (PMID: 12381664). *Zic3*^{-/-} mice are viable, but exhibit defects of the cardiac and central nervous systems. We now describe these phenotypes in the discussion section.

4) Is there a minimal threshold of Xist activity that allows proper XCI? It seems to me that authors are able to address this question using their RE-TF interplay. It will significantly add to the mechanistical understanding of the dual Xist activation (why and when).

We thank the reviewer for their suggestion, prompting us to study the relationship between Xist levels and XCI in more detail. Using our CasTuner system we titrated Xist during differentiation and assess chromosome-wide gene silencing dynamics. Our results do not reveal a clear threshold for Xist activity. Instead, silencing efficiency increases gradually with higher Xist levels and then saturates. The majority of wildtype cells exhibit Xist levels towards saturation, cells of the *Ftx-Xert* deletion line exhibit lower Xist levels associated with strongly reduced silencing. This new data set is presented in Fig. 6j-o and Extended Data Fig. 11.

5) As authors commence their intro focusing on a general gene regulatory question, I wonder if distal/ proximal function of Zic3, Nfrkb, Otx2, Foxd3 is something that these factors exhibit in general (i.e. their other targets) or happens only during XCI and for Xist. Their non-X targets would be interesting to explore/provide in the context of these observations. Similarly, how scalable are the observations to other ncRNAs or coding genes? Xist is a very specific locus like no other in a mammalian genome, thus, it's regulation might not be easily translatable to other loci.

We agree with the reviewer that the generalizability of our observations at the Xist locus is an intriguing question. However, it is fairly challenging to answer. We therefore feel that it is beyond the scope of this manuscript. Instead, we have added a part to the discussion section formulating questions arising from our manuscript for future studies. We also discuss the question to what extent Xist can be a model to understand general principles of transcriptional regulation.

Minor Comments

6. In the undifferentiated state, where Oct4 acts as a repressor, its effects seem much stronger than in its later role as an activator. The authors should address this difference, considering both the direct and indirect effects of Oct4 are proposed.

Since Xist is normally expressed at very low levels in undifferentiated cells, the Oct4 knock-down effects are large in terms of fold change. With respect to absolute expression differences are lower in 2iSL compared differentiated cells and of similar magnitude in SL (see figure below).

Rebuttal Fig.1: Effect of Oct4 knockdown on Xist and Oct4 expression, assessed via real-time quantitative PCR (RT-qPCR). The experiment was performed in three biological replicates. Significance between control and knockdown is marked with a black asterisk (*unpaired t-test*, $p \leq 0.05$).

7. It would be helpful to explain how the FIREWACH system works, unfortunately Fig 3a is not helpful in that sense as it is unclear what binds to what and upon which signal.

We have modified Fig. 4a as well as the associated text to improve clarity.

8. How do authors decide to use 2iL withdrawal as their experimental model for differentiation and not RA or Fgf2/AcA protocols? It should be explained.

In our hands the 2iL withdrawal system enables efficient high-level Xist upregulation. We have therefore characterized it extensively in a series of prior studies and also used it in the present manuscript. Through integration with embryo data we show that it represents the *in vivo* process well, which we now also point out at the beginning of the results section.

9. Was the Xist detection range (by FISH or other means) comparable in the two screens?

To compare the Xist detection range during the FlowFISH staining, we calculated the difference between the \log_{10} -transformed fluorescence values of the Xist^{high} and Xist^{low} sorting gates across both screens. Indeed, the detection range was consistent across all four samples. We included the data in Extended Data Fig. 3b.

10. If the second screen is performed after longer time, wouldn't it deliver different group of regulators (mainly those acting in later stages of differentiation)? I might have missed it but it would be good to explain in case that's not the expectation.

Both screens are performed at day 2 of differentiation. We would therefore expect to find similar regulators. What was altered was the adaptation time from SL to 2iSL conditions, essentially to increase robustness of the results. However, the strong correlation between the screen results shows that the findings are not affected by the adaption time. We have simplified the results section and only discuss this in the methods and in Supplementary Note 1 to avoid confusion.

11. In Figure 2 it seems like there's a panel missing with the actual 2nd screen results. I see it in the supplement, but it would be great to have it in the main. Can authors provide the tracks from sorting?

Following the reviewer's suggestion, we have now combined both screens in Fig. 1. A figure describing the sorting during the TFi screen is now provided in Extended Data Fig. 1g.

12. Out of curiosity, wouldn't RNA decay mechanisms that regulate stability of Xist affect results/conclusions of the screens? It seems like an important mechanism that is linked with the need to boost Xist expression at certain differentiation time-point.

By design, pooled CRISPR screens are typically agnostic to the underlying regulatory mechanisms. By restricting our library design to genes with reported transcription factor activity, we aim to primarily detect transcriptional regulators of Xist. That said, we cannot rule out the possibility that positive hits may impact Xist expression levels via alternative mechanisms, including altering the stability of the Xist RNA. We now discuss this point in Supplementary Note 2.